# A comparative flood damage and risk impact assessment of land use changes

Karen Gabriels[1], Patrick Willems[1], Jos Van Orshoven[2]

[1]Department of Earth and Environmental Sciences, KU Leuven, Leuven, 3001, Belgium
[2]Department of Civil Engineering, KU Leuven, Leuven, 3001, Belgium

*Correspondence to*: Karen Gabriels (karen.gabriels@kuleuven.be)

**Abstract.** Sustainable flood risk management encompasses the implementation of nature-based solutions to mitigate flood risk. These measures include the establishment of land use types with a high (e.g. forest patches) or low (e.g. sealed surfaces) water retention and infiltration capacity at strategic locations in the catchment. This paper presents an approach for assessing the relative impact of such land use changes on economic flood damages and associated risk. This spatially explicit approach integrates a reference situation, a flood damage model and a rainfall-runoff model, considering runoff re-infiltration and propagation, to determine relative flood risk mitigation or increment related to the implementation of land use change scenarios. The applicability of the framework is illustrated for a 4800 ha undulating catchment in the region of Flanders, Belgium by assessing afforestation of 187.5 ha (3.9%), located mainly in the valleys, and sealing of 187.5 ha, situated mainly at higher elevations. These scenarios result in a risk reduction of 57% (€ 100K) for the afforestation scenario and a risk increment of <1% (~ 500 €) for the sealing scenario.

## 1 Introduction

River flooding is a natural process, but also poses a significant socioeconomic hazard, causing human distress and damage to properties and infrastructure. In Europe, floods caused approximately 147 billion euros economic damages between 1980 and 2019 (EEA, 2021). Moreover, the economic losses associated with flood events have been on the increase in the past decades (since 1970), partly due to changing weather patterns (IPCC, 2014), but mainly driven by socioeconomic developments such as population growth, increasing wealth and ongoing urbanization in flood prone areas (Barredo, 2009; Bouwer, 2011; Koks et al., 2014). The increasing flood losses prompted a shift in flood management in Europe from a flood prevention policy to flood risk management policy (EEA, 2017), as detailed in the European Flood Directive (Directive 2007/60/EC, 2007). Flood risk management aims at minimizing flood risk, which is defined by the probability of a flood event and its potential negative consequences, also termed flood damages. Flood risk is thus an expression of the expected flood damages over a certain period of time, e.g. the expected annual damages (Bubeck et al., 2011; Grossi and Kunreuther, 2005; Merz et al., 2010; de Moel et al., 2015).

The first step in the general approach for flood risk assessments (de Moel et al., 2015) is to derive indicators of flood hazard,

i.e. the probability and intensity of floods, from flood maps. These flood maps typically represent the flood extent and water depth of hypothetical flood events with different probabilities of occurrence (de Moel et al., 2009). Next, the corresponding flood damages are determined in flood damage models, which relate the flood hazard characteristics, established in the flood maps, to the vulnerability to flooding of the exposed assets, i.e. the ecosystems, people and properties at risk. Finally, the flood risk is determined by combining the flood damages caused by flood events with different return periods in a weighted

summation.

Flood damage entails all negative, harmful impacts of floods on society, economy and the environment. Generally, direct and indirect damages are distinguished. Direct flood damage occurs at the time of flooding through the physical contact of the exposed elements with flood waters, while indirect flood damage relates to the induced losses as a result of flooding, e.g.

production losses (Merz et al., 2010). A second distinction is made between tangible and intangible damages: tangible damages can easily be expressed in monetary values, whereas intangible damages encompass damage inflicted on assets of which the financial value is more difficult to assess. Examples of direct, tangible flood damage include damage to buildings and household effects, whereas direct, intangible damages encompass loss of life and damage to cultural heritage. Indirect, tangible flood damages are, for instance, the induced production losses of companies situated outside the flooded area, while indirect

and intangible damage entails the psychological impact of exposure to flooding (Merz et al., 2010; Messner and Meyer, 2006). Flood risk analyses often only comprise an assessment of tangible flood damages, which are easier and more reliable to estimate than intangible flood damages (Merz et al., 2010). The vulnerability to flooding of assets is described by damage functions, providing a link between the valuation of the assets exposed to the flood and the corresponding flood hazard characteristics, established in the flood maps. Most often, damage functions are included in flood damage models in the form

of depth-damage curves, detailing the impact of water depth on the value of the assets exposed to flooding (Gerl et al., 2016).

An example of a flood risk analysis tool is LATIS, developed in Flanders, Belgium based on the damage model of Vanneuville et al. (2006). The economic damage assessment in LATIS considers direct and indirect flood damages (Beullens et al., 2017; Kellens et al., 2013; VMM, 2018). The depth-damage functions implemented in LATIS are expert-based. Due to the lack of

consistent, complete and spatially distributed data on insured flood damages in Flanders, these depth-damage curves are mostly derived from enquiries conducted in the Netherlands and the United Kingdom (UK), in addition to a limited comparison with recorded damages in the Belgian disaster fund (Vanneuville et al., 2006). In the Netherlands, flood risk frameworks were implemented by Ward, de Moel, & Aerts (2011) and de Moel, van Vliet, & Aerts (2014) based on the Damage Scanner model, which assesses direct and indirect economic flood damages. The depth-damage functions in the Damage Scanner are based on

expert-knowledge and available damage statistics (Klijn et al., 2007). In the UK, flood risk assessments (e.g. Hall, Sayers, & Dawson, 2005) commonly implement the damage model presented in Penning-Rowsell et al. (2005), assessing both direct and

indirect economic damages. Expert-based damage functions are implemented, which assess flood damage considering both water depth and flood duration.

By explicitly taking into account potential flood damages, these risk assessments identify people and assets at risk of flooding, which in turn is a basis for the determination of flood insurance premiums (Grossi and Kunreuther, 2005; Merz et al., 2010) and to evaluate the effect and efficiency of flood mitigation measures (Koks et al., 2014; de Moel et al., 2014). As flood risk management has continued to evolve into an integrated, system-wide approach, flood mitigation measures are increasingly incorporating nature-based solutions (EEA, 2015; Sayers et al., 2015; SEPA, 2016). Such measures include the preservation

and establishment of natural ecosystems at strategic locations in catchments, since vegetated systems have the capacity to influence the hydrology of small- to medium-sized catchments by enhancing water retention and infiltration (Bronstert et al., 2002; Peel, 2009). Conversely, the process of sealing soil surfaces for urbanization, e.g. with concrete surfaces, makes these surfaces impermeable and prevent water to infiltrate into the soil, thus decreasing the potential for water storage and increasing the fraction of rapid surface runoff accumulating in downstream areas (Lin et al., 2007; Miller et al., 2014; Poelmans et al.,

2011). Consequently, land use systems have the capacity to either mitigate or exacerbate flood damage and risk downstream. Based on this rationale, we present a spatially explicit, comparative flood risk assessment framework to evaluate land use changes as flood mitigation measures. This framework compares direct, tangible economic flood damages and the associated risk before and after specific land use change scenarios, whereby the original land use serves as a baseline scenario.

The methodological procedure of the comparative risk framework is first elaborated, after which an application of this framework is presented on a case study in the Maarkebeek basin in Flanders, Belgium. Flood extents in Flanders have been recorded in a geospatial flood archive outlining the maximum extent of flooded zones from 1988 to 2016 (AGIV and VMM, 2017; Van Orshoven, 2001). Using a flood damage model based on the depth-damage curves of LATIS, flood damages were assessed from several flood events occurring in the Maarkebeek basin between 2000 and 2016, of which the extent is recorded

in the geospatial flood archive. The overall flood risk was determined by combining the flood damages of these events with their respective probability of occurrence. Next, two land use change scenarios were taken into consideration in this case study, namely an afforestation scenario and soil sealing scenario. Subsequently, the corresponding hydrological impact of these land use change scenarios was calculated by a spatially explicit rainfall-runoff (RR) model, calculating the runoff volume accumulated in each pixel after a rainfall event. Based on the accumulated runoff volume after land use changes, the altered

flood extents and water depths were derived, and the corresponding flood damages and flood risk were calculated. Finally, flood damage and risk before and after land use changes were compared to provide the relative impact of the considered land use changes on the downstream flooded areas.

## 2 Material and Methods

### 2.1 Comparative flood damage and risk assessment

The framework determining the spatially explicit, relative flood damage and risk impact of land use changes is visualized in Figure 1. First, flood depths and volumes are derived from observed, rasterized flood extents for multiple return periods before any implementation of land use changes. Next, the hydrological impact of a land use change scenario is determined by a spatially explicit RR-model, which calculates the volume of runoff accumulated in each pixel (Gabriels et al., 2021). Consequently, an empirical relationship between observed flood volumes and modeled runoff volume accumulation is

established to determine the flood volumes after land use changes. Based on these modeled flood volumes, a Digital Elevation Model (DEM) is progressively filled and corresponding water depths are thus determined. The water depths before and after land use change are then combined with socio-economic information in a flood damage model to determine the corresponding flood damages. In this flood damage model, only direct, economic flood damages were taken into consideration and expressed as a monetary values. The difference between the flood damage datasets before and after land use change is defined as the

relative flood damage impact of the land use changes. In order to evaluate the overall flood risk impact, the flood damages of several flood events with different probabilities are combined.

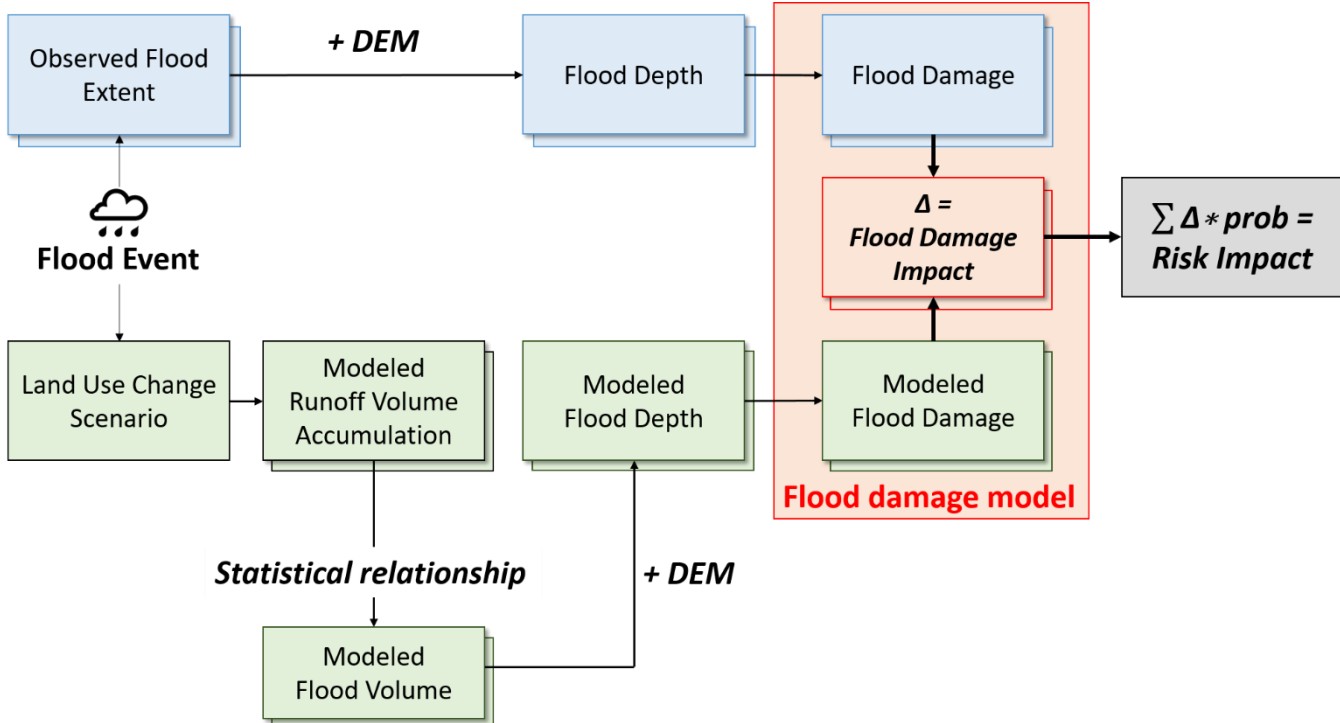

**Figure 1: Framework determining the overall flood damage and risk impact of land use changes.**

### 2.1.1 Flood depth and volume calculations before and after land use changes

Rasterized flood extents, related to a specific flood event, are first combined with a DEM to derive the water depth in each of the flooded pixels. This water depth is determined by fitting a linear, least-squares plane representing the water level elevation across each flood extent based on the elevation of the pixels bordering the flood extents and the pixels representing the river banks. The water elevation is then corrected for each pixel, by averaging this elevation with the water level determined by a local, linear interpolation only based on the nearest flood border pixels. Finally, the water depth is calculated per pixel by

subtracting the DEM from the water level. Consequently, the flooded volume in each pixel is calculated by multiplying the water depth with each pixel's area, determined by its resolution.

Next, the rainfall and antecedent soil moisture condition of each flood event together with the land use in the watershed are modeled by the RR-model to determine the runoff volume accumulated in each pixel of the basin during the flood event.

Runoff volume is determined using the empirical, event-based Curve Number (CN) method. The CN method determines runoff based on rainfall volumes and the CN parameter, which depends on land use and soil information. Prior to the runoff calculation, the CN parameter in the RR-model is adjusted to reflect soil moisture conditions antecedent to the rainfall event following the method implemented by Neitsch et al. (2011), whereby a higher soil moisture leads to an increase in CN. The RR-model then propagates the runoff through the watershed, thereby continuously assessing downstream re-infiltration using

the Manning's equation. Consequently, spatial connectivity between pixels is taken into account. Further details on this RR-model can be found in Gabriels et al. (2021). The hydrological impact of land use changes is simulated using the same RR-model by adjusting the model parameters related to land use, i.e. the CN value and Manning's roughness coefficient (Gabriels et al., 2022).

In order to relate the modeled runoff volume accumulation with flood volume, an empirical function is fitted through these two variables. Analogously to the relationship found by Mediero, Jiménez-Álvarez, & Garrote (2010) between flood peak discharge and flood volume, a linear relationship is determined in the log-log space between the total flood volume *Vol* in the flood extent *j* and the accumulated runoff volume *Q* at the flood extent's outlet, i.e. the most downstream pixel in each extent:

$$Vol_j = 10^a * Q_j^b, \tag{1}$$

with *a* and *b* respectively the intercept and coefficient of the linear relationship. Using this correlation, the simulated accumulated runoff volume resulting from the land use change scenarios can then be expressed as a flood volume. Based on this simulated flood volume, the altered flood extent and corresponding water depth is determined by progressively filling the DEM covering the original flood extent, similar to the simple, conceptual "bathtub" method (Teng et al., 2015).

### 2.1.2 Flood damage model

Flood damages before and after land use changes are determined for each pixel by combining the derived water depth datasets with a flood damage model. The flood damage model estimates the direct economic damages per land use class based on depth-damage curves, relating the water depth with a damage factor $\alpha$ (Koks et al., 2014). The total effective flood damage $D$ in each pixel is then calculated by multiplying this damage factor $\alpha$ with the maximum possible flood damage $D_{max}$ (€/m² or €/m for road infrastructure), summed over the different land use classes in the pixel:

$$D = \sum \alpha * D_{max}, \tag{2}$$

The depth-damage curves implemented in the flood damage model are the expert based functions from Vanneuville et al. (2006). They are provided in Figure 2 for the different land use classes.

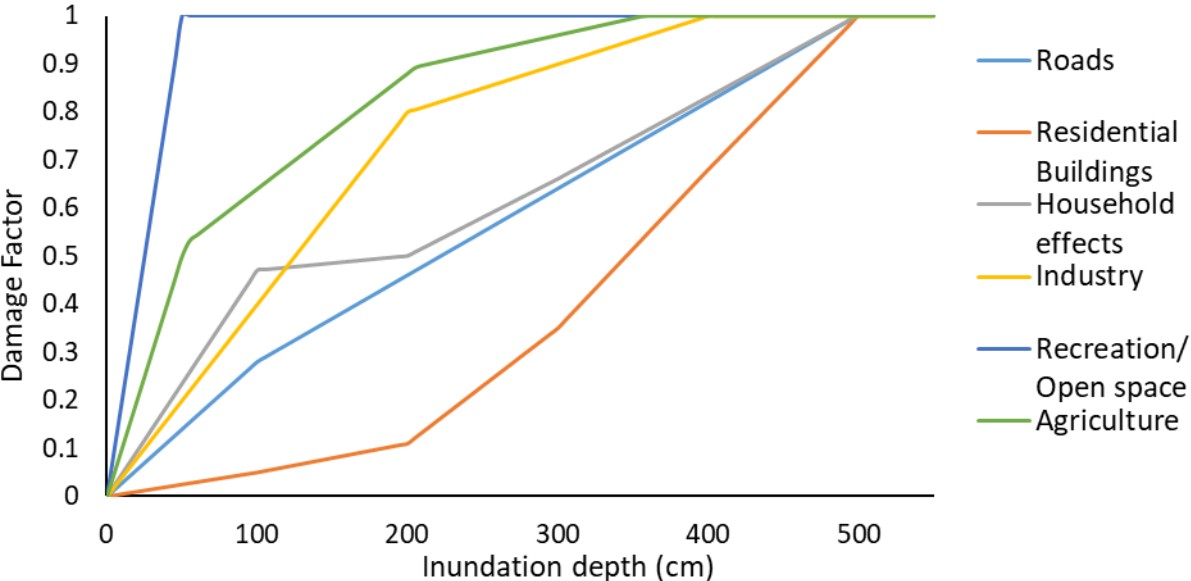

**Figure 2: The flood damage curves depicting the relationship between the inundation depth (cm) and the damage factor (Vanneuville**
**et al., 2006).**

The maximum damage values implemented in the flood damage model are provided in Table 1 per land use class. These amounts were established based on the replacement values implemented in the LATIS tool (Beullens et al., 2017; Vanneuville et al., 2006) and in Koks et al. (2014); these values were not adjusted to the price level in a specific year. These maximum damage estimates were also not spatially differentiated and thus assumed valid for Flanders, with the exception of the

maximum damage to residential buildings. Similar to the method applied in LATIS, the maximum flood damage to residential buildings was derived from socio-economic data regarding the median residential housing price in a municipality divided by its average housing surface area. The maximum damage to household effects was estimated at 30% of the damage to residential buildings, while damage to residential open space, including damage to garden houses, was set at € 1/m² (Kellens et al., 2013). The maximum damage to industrial buildings was estimated at a unity price of € 700/m² (Koks et al., 2014), while maximum

damage to industrial open spaces, including industrial installations and supplies, was estimated € 100/m² (Kellens et al., 2013; Vanneuville et al., 2006). Maximum damage to road infrastructure is dependent on the type of road, ranging between € 41/m for dirt roads and € 1374/m for highways, as determined by Beullens et al. (2017). The maximum damage to arable land mainly relates to losses in crop production. Though LATIS distinguishes maximum damage estimates for different crops, the loss of crop production in the flood damage model of the flood risk assessment framework was to a constant value € 0.5/m² for all arable land, while the maximum damage to grasslands, including pastures and meadows, was estimated at € 0.08/m². Damage to natural areas, such as forests, was set to € 0/m² (Kellens et al., 2013; Vanneuville et al., 2006).

**Table 1: The maximum damage values as implemented in the flood damage model and derived from (Beullens et al., 2017), (Koks et al., 2014) and (Vanneuville et al., 2006).**

| Land use class | Damage Function | Maximum damage |
|---|---|---|
| Residential Buildings | Residential Buildings | *Housing price* /m² |
| Residential Household effects | Household effects | 30% of *Housing* price /m² |
| Industrial Building | Industry | € 700 /m² |
| Open space | Recreation/Open Space | € 1 /m² (residential) – € 100 /m² (industrial) |
| Roads | Roads | € 41–1374 /m |
| Arable land | Agriculture | € 0.5 /m² |
| Grassland | Agriculture | € 0.08/m² |

**2.1.3 Risk calculations**

The damage datasets derived from the flood damage model for flood events with different probabilities or return periods are combined to assess the change in flood risk from the implemented land use changes. Flood risk $R$ is calculated by adding the flood damages $D$ of all flood events under consideration, thereby weighing these damages according to their corresponding return period $i$. This weighted summation takes into account the damages of events with lower return periods to avoid double counting damages of these more frequent events. This is mathematically expressed as (Kellens et al., 2013):

$$R = \sum_{i=1}^{n} \frac{1}{i} * (D_i - D_{i-1}), \tag{3}$$

Since only a limited number of return periods are assessed, a linear interpolation is performed between two return periods $x$ and $p$, which can be expressed as (Deckers et al., 2009; Vanneuville et al., 2003):

$$R = \sum_{i=x} \left( \frac{\frac{1}{p+1} + \cdots + \frac{1}{x}}{x - p} \right) * (D_x - D_p), \tag{4}$$

Where $p$ is a smaller return period than $x$.

**2.2 Case study**

**2.2.1 Baseline flood damage and risk assessment of observed flood events**

The framework was implemented in a case study in the catchment of the Maarkebeek (48 km²), situated in the Upper Scheldt basin in Flanders, Belgium. This is a mostly agricultural area, dominated by arable land. Approximately 10% of the catchment is urbanized and about an equal area is afforested. A general depiction of the land use in the area is provided in Figure 3 with

a resolution of 50 m. The RR-model was validated for the Maarkebeek catchment for 165 rainfall events, resulting in a Nash-Sutcliffe Efficiency (NSE) of 0.57 (Nash and Sutcliffe, 1970). As such, the RR-model was deemed sufficiently accurate to compare the hydrological impacts of land use changes (Gabriels et al., 2021). The land use changes are consequently modelled with a resolution of 50 m.

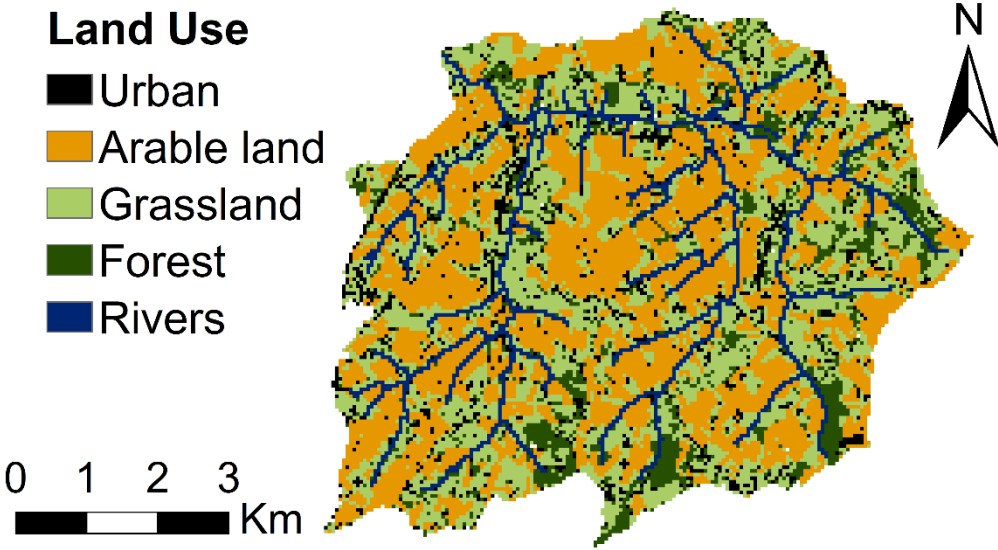

**Figure 3: General land use in the Maarkebeek catchment, based on the land use dataset of 2012 (AGIV, 2016).**

Flood damage and risk were assessed from observed flood extents derived from the geospatial flood archive. This geospatial flood archive details the maximum extent of flooded areas in Flanders for flood events between 1988 and 2016 (AGIV and VMM, 2017). Eight flood events were registered in the geospatial flood archive for the Maarkebeek catchment, namely one flood event in each of 1993, 1995, 1998, 1999, 2003 and 2010 and two flood events in 2002. Since the rainfall dataset ranges

from 2000 to 2012, the risk assessment was performed on the four flood events observed after 2000, i.e. two flood events taking place in 2002 (19-27/02/2002 and 19-21/08/2002), one flood event in 2003 (1-3/01/2003) and one flood event in 2010 (11-15/11/2010). The extents of the flooded areas during these events are visualized in Figure 4: one flood extent was registered in each event in 2002, while respectively three and eight separate flood extents were observed in 2003 and 2010. The flood extents in February and August 2002 do not overlap and are depicted in the same figure. Flood extents situated partially or

completely outside the Maarkebeek catchment were not taken into consideration.

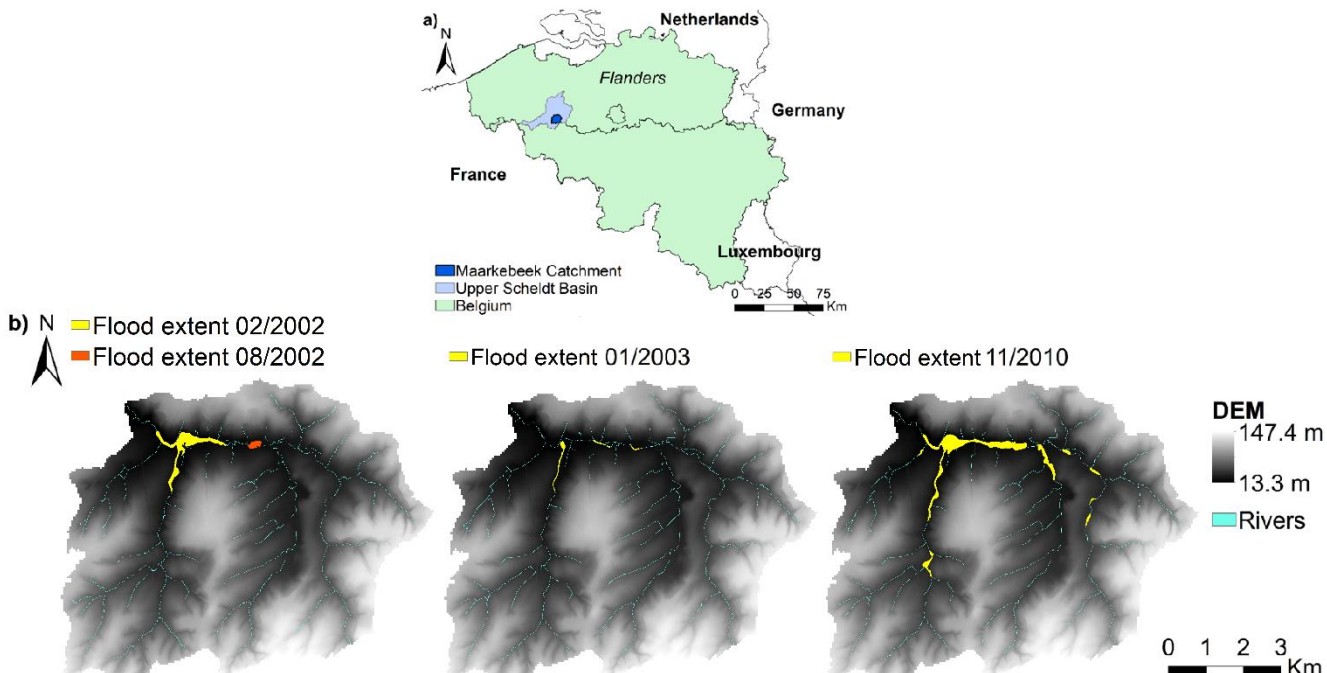

**Figure 4: Extents of flooded areas in the Maarkebeek basin as recorded in the geospatial flood archive for the 2000–2016 period (AGIV et al., 2006; AGIV and VMM, 2017). One flood extent was recorded in resp. February and August 2002, these extents do not overlap and are depicted in the same figure. Resp. three and eight flood extents were recorded in the flood events in 2003 and 2010.**

For each of these flood events, the water depths in the corresponding flood extents were first determined. Consequently, the flood extents were rasterized with a resolution of 5 m and then combined with a DEM to fit a linear plane, as described above, to determine the water level and associated water depth in each pixel (AGIV et al., 2006). Based on these water depths, the flood damages were assessed on a per-pixel basis using the flood damage model. Socio-economic information and land use datasets regarding the land use classes in Table 1 were collected to determine the maximum flood damage in each pixel. The

maximum damage to residential buildings was determined by combining the median residential housing price in 2002, 2003 and 2010 in the municipalities situated in the Maarkebeek subcatchment (Oudenaarde, Ronse, Brakel, Horebeke and Maarkedal) (Statbel, 2019) with the number of residences and their total surface area in the municipalities, which was derived from a high resolution dataset outlining building footprints (AGIV, 2020). These residential damages ranged from € 439/m² to € 703/m² in 2002, from € 492/m² to € 745/m² in 2003, and from € 903/m² to € 1524/m² in 2010. Road infrastructure in the

catchment was derived from the road register (AGIV and NGI, 2020). According to the industrial parcel dataset (VLAIO and AGIV, 2020), no industrial areas were flooded during these four events. The non-residential and non-industrial land use classes in Table 1, i.e. arable land, grassland and open space, were derived from the land use dataset from 2012 with a resolution of 5 m (AGIV, 2016).

Next, the flood risk corresponding to these flood damages was determined according to Eq. (4), for which the return period of

each flood events was empirically estimated by applying the Weibull formula on an analysis of the annual maximum discharge

(Chow et al., 1988), based on discharge data from 1973 to 2019 of the Maarkebeek river (VMM et al., 2020). In this analysis, 45 annual maxima were included, as data from 2016 and 2017 was incomplete. This analysis estimated the return period of the 2010 flood event at 46 years, since the highest discharge of the time series was recorded during this event. The flood event in 2003 had a return period of 3 years, while the February and August 2002 flood events had return periods of, respectively, 11 and 1 year(s). Implementing these values in Eq. (4) results in the following formula to assess the flood risk $R$ based on the damages $D$ corresponding to these events:

$$R = 0.58 * D_1 + 0.27 * D_3 + 0.11 * D_{11} + 0.04 * D_{46}, \tag{5}$$

### 2.2.2 Comparative flood damage and risk assessment of land use changes

After determining the observed flood damage and corresponding flood risk over all four flood events, the relative impact to this base-line was assessed for two types of land use changes, afforestation and soil sealing. First, two land use change scenarios were derived through a raster-based optimization procedure that identifies locations for the considered land use change having maximal impact on the flood volume. This procedure ranks pixels based on (i) where in the upstream area of the flooded zones afforestation maximally reduces the runoff accumulation in these zones, and (ii) where upstream soil sealing would lead to the smallest increase in runoff accumulation, in each of the flood extents of all considered flood events. Land use changes are simulated through an adjustment of the RR-model CN parameter and the Manning's runoff coefficient, with afforestation and sealing resp. leading to a decrease or increase in runoff volume and velocity. By simulating the land use changes with the RR-model, spatial connectivity between pixels is taken into account. More details regarding the optimization procedure can be found in Gabriels et al. (2022). In each of the two flood events in 2002, only one flood extent was observed; the most downstream pixel in this extent, i.e. the outlet, was consequently used as point of interest (POI) in the optimization and pixels were ranked based on the change in runoff volume accumulation at this POI. In the flood events in 2003 and 2010, respectively three and eight flood extents were observed. These extents' outlets were considered the POIs in the optimization and the pixels were ranked based on the combined changes in runoff accumulation at these pixels, weighted according to the observed flood damages in each flood extent. The four optimization results, one for each of the flood events, were summed to obtain one ranking for each land use change, thereby weighting the standardized pixel ranks according to the flood hazard, i.e. as the corresponding flood damages are weighted in Eq. (5). Based on this final priority rank, the top 750 pixels, representing 187.5 ha or approximately 4% of the study area, were selected, for both the afforestation and the sealing scenario. Figure 5 depicts the resulting afforestation and soil sealing scenarios. The pixels to be afforested are mostly located along the rivers, whereas pixels to be sealed are located in the more elevated parts of the catchment, away from the rivers and situated near forest patches. The optimization procedure therefore leads to the hypothesis that building in the uplands leads to a lesser increase of flood hazard than building in the lowlands, and that afforestation of the riparian zones leads to a larger reduction of flood hazard than afforestation of the uplands. These findings are consistent with the results of Yeo and Guldmann (2010). The selected

pixels are mainly situated in the eastern part of the catchment, upstream from most flood extents: these pixels have higher ranks as land use changes in these pixels will have an impact on more flood extents.

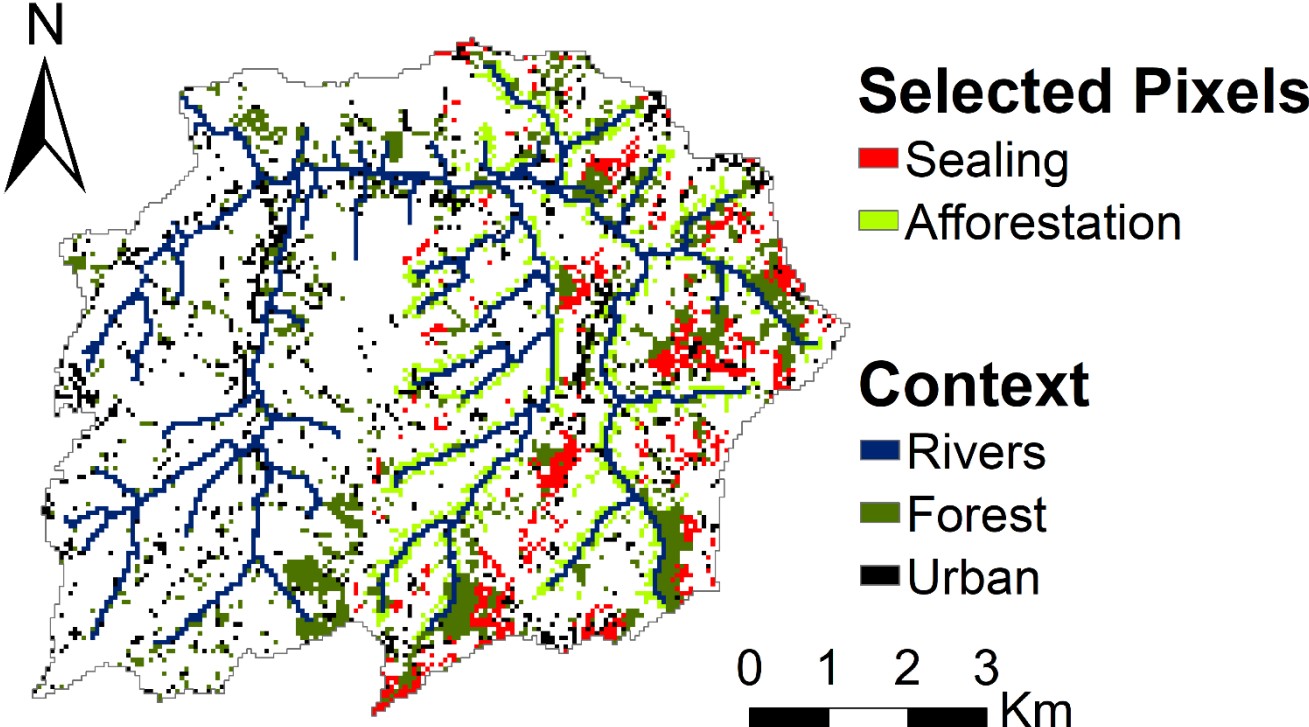

**Figure 5: Locations of the pixels selected for land use change implementation, i.e. the 750 priority pixels (187.5 ha), for both the afforestation and soil sealing scenarios.**

Next, the runoff volume accumulation $Q$ of each flood event was modeled with a resolution of 50 m, based on the land use dataset from 2012 (AGIV, 2016) and meteorological information from the Royal Meteorological Institute and the Flanders Environment Agency (Van Opstal et al., 2014). Subsequently, the empirical relationship, analogously to Eq. (1), between the modeled runoff volume accumulation $Q$ at the corresponding extent's outlet and the derived flood volumes $Vol$ of the thirteen observed flood extents was fitted with an adjusted R² of 0.76:

$$Vol = 10^{-6.32} * Q^{1.9}, \tag{6}$$

This relationship was used to determine the flood volume before and after implementing the land use change scenarios based on the corresponding modeled accumulated runoff volume. Based on these flood volumes, the DEM of the corresponding flood extents were filled to determine the water depths with a resolution of 5 m. The flood damage and risk assessment was then implemented on these water depths before and after land use changes; and based on the difference between flood damage and risk, the relative impact of these land use changes was assessed.

## 3 Results

### 3.1 Baseline flood damage and risk assessment of observed flood events

Statistics regarding the flood events are provided in Table 2, which details the flooded area, volume and damage for each flood extent in each of the four flood events, as well as the modeled accumulated runoff volume at each extent's outlet and the corresponding, modeled flood volumes derived with Eq. (6). Figure 6 depicts the relationship, with an adjusted $R^2$ of 0.76, between the observed and modeled flood volumes.

**Table 2: Overview of the flooded area (ha), total observed flood volume (m³), resulting flood damages (€), runoff volume accumulation at the flood extents' outlet (m³) and total modeled flood volume (m³) for each of the four observed flood events and their corresponding flood extents.**

| Event | Extent | Flood Area (ha) | Flood Vol. (m³) | Damages (€) | Runoff Vol. Acc. (m³) | Flood Vol. (m³) *Modeled* |
|---|---|---|---|---|---|---|
| *02/2002* | | 31 | 153 321 | 566 667 | 1 143 815 | 156 840 |
| *08/2002* | | 4.2 | 14 699 | 27 515 | 199 406 | 5667 |
| *2003* | Extent 1 | 4.3 | 24 698 | 49 693 | 295 820 | 11 994 |
| | Extent 2 | 1.0 | 6032 | 68 405 | 282 365 | 10 978 |
| | Extent 3 | 0.7 | 4003 | 21 552 | 258 176 | 9260 |
| | **Total** | **6** | **34 733** | **139 650** | | |
| *2010* | Extent 1 | 43.2 | 243 407 | 827 122 | 1 504 926 | 264 226 |
| | Extent 2 | 0.7 | 2814 | 60 594 | 136 299 | 2749 |
| | Extent 3 | 7.2 | 55 990 | 366 219 | 433 442 | 24 794 |
| | Extent 4 | 1.4 | 6069 | 4414 | 335 090 | 15 201 |
| | Extent 5 | 1.4 | 7331 | 11 382 | 303 962 | 12 629 |
| | Extent 6 | 0.5 | 2848 | 43 835 | 199 504 | 5672 |
| | Extent 7 | 1.2 | 6923 | 100 976 | 188 777 | 5107 |
| | Extent 8 | 3.7 | 35 724 | 141 813 | 331 361 | 14 881 |
| | **Total** | **59.3** | **361 106** | **1 556 355** | | |

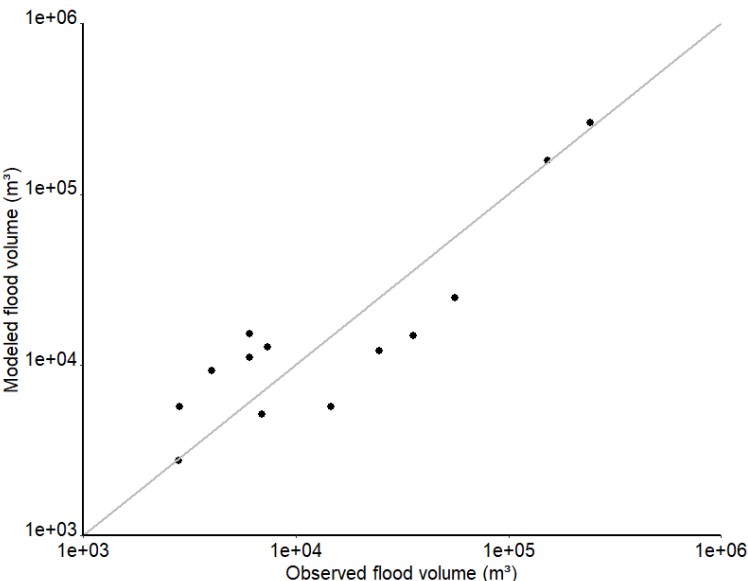

**Figure 6: Scatterplot of the flood volumes derived from the observed flood extents (Observed flood volume, m³) and the flood volumes as modeled by Eq. (6) (Modeled flood volume, m³), with an adjusted R² of 0.76 and a relative RMSE of 0.3.**

The water depth and corresponding flood damage datasets are shown per pixel in Figure 7. The highest water depths were obtained in river pixels and pixels bordering the river. The flood damages are highly localized, with the highest damages inflicted in built-up pixels containing roads and residential buildings. The maximum flood damage in a pixel (25 m²) was € 5493 or approximately € 220/m². The total flood damage was respectively € 566 667, € 27 515, € 139 650 and € 1 556 355 for the flood events in February 2002, in August 2002, in January 2003 and in November 2010 (Table 2). During these four

flood events, a total flood damage of € 2 290 187 was inflicted in the Maarkebeek catchment. The flood damage datasets were combined according to Eq. (5) to determine flood risk or the expected annual damages in each pixel, as depicted in Figure 8. Similarly to flood damage, flood risk is highly localized and highest (€ 1265/year in a pixel or € 50.6/year/m²) in repeatedly flooded, built-up pixels. The total flood risk derived from the four flood events in the Maarkebeek catchment equals € 178 252/year.

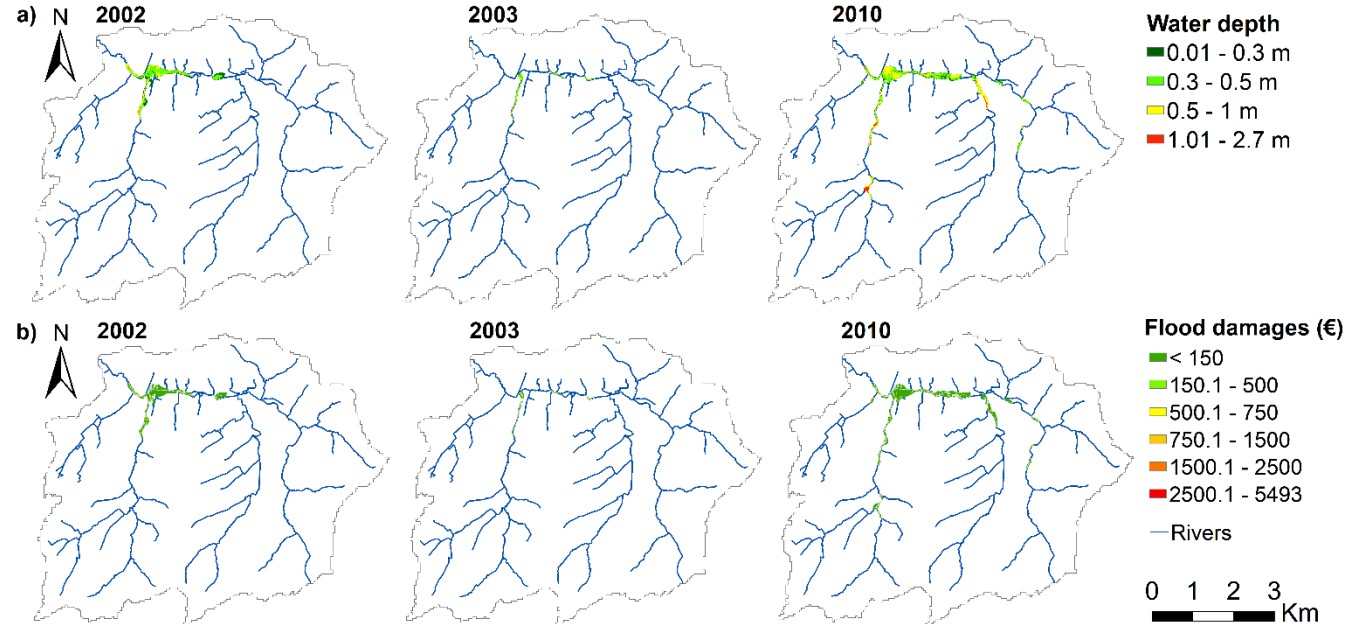

**Figure 7: (a) The inundation depth (m) and (b) the corresponding flood damage (€) per pixel (5m X 5m resolution) derived from the flood damage model in the Maarkebeek catchment resulting from the observed flood events. The total flood damage was respectively € 566 667, € 27 515, € 139 650 and € 1 556 355 for the flood events in February 2002, in August 2002, in January 2003 and in November 2010.**


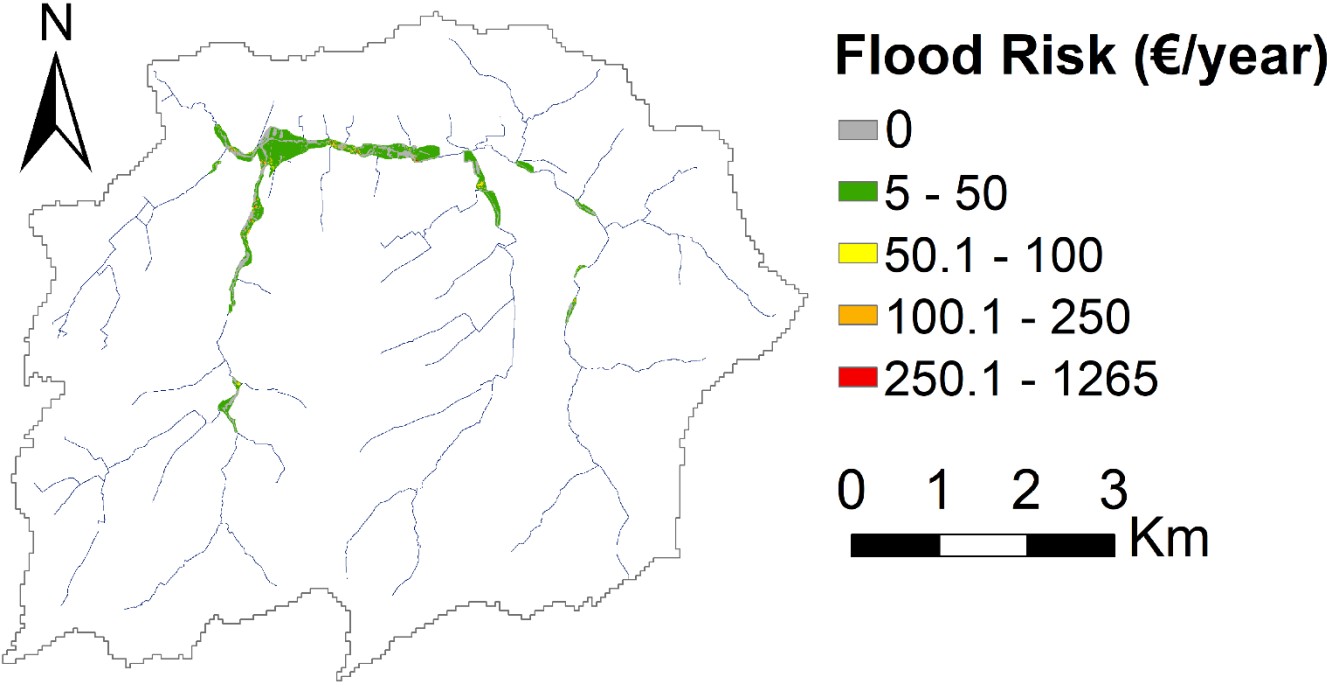


**Figure 8: Flood risk, expressed as expected annual damages (€/year) in each pixel (5m X 5m resolution), in the Maarkebeek catchment based on the four observed flood events. Flood risk is highly localized and highest (€ 1265/year or € 50.6/year/m²) in only**

**a few, built-up pixels which were repeatedly flooded. The total flood risk derived from the four flood events in the Maarkebeek catchment equals € 178 252/year.**

## 3.2 Comparative flood damage and risk assessment of land use changes

Figure 9 depicts, for each flooded pixel, the relative decrease in flood damages after afforestation of the 750 highest impact pixels, i.e. the relative flood damage mitigation, and the relative flood damage increment after implementing the sealing scenario (750 lowest impact pixels). This information is summarized in Table 3 for every flood extent and for each of the flood events. The relative flood damage mitigation after implementing the afforestation scheme was -41.4% and -97.3% in respectively February and August 2002, -91.5% in 2003 and -39.3% in 2010. The high damage reduction in the flood event of 2003 is explained by the flood volumes in the two most upstream, smaller flood extents in this event being reduced to nearly zero (Table 3). The flood damage reduction is highest where the water depth is reduced in built-up urban areas. For the entire Maarkebeek catchment, the afforestation scenario reduced flood damages with 44.7%, which equals an absolute reduction of € 1 023 714. The relative damage increment after sealing the 750 least runoff incurring pixels equals 1.1% and 2.8% in respectively February and August 2002, 0.01% in 2003 and 1.9% in 2010. The damage increase is mostly due to new pixels being flooded, however, it is limited due to the unbuilt nature of these areas, as the soil sealing took place in the uphill areas of the catchment, away from the rivers and flooded areas. Total flood damages in the Maarkebeek catchment increased with 1.5%, which resulted in an increase in total flood damage after soil sealing of € 34 353.

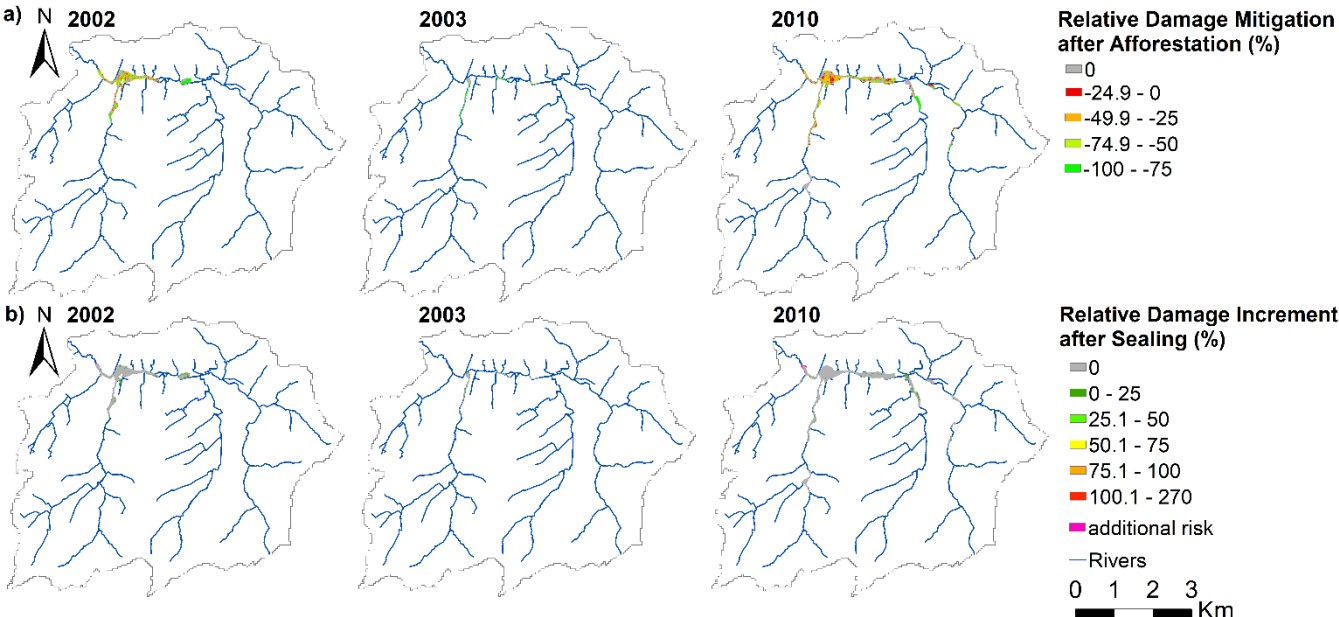

**Figure 9: The relative impact in flood damages (%) after (a) implementing the afforestation scenario, resulting in a relative damage mitigation, and after (b) implementing the soil sealing scenario, resulting in a relative flood damage increment. New areas being flooded after soil sealing are depicted as 'additional damage', though these areas are limited to a few pixels bordering the river or existing flood extents.**

**Table 3: Relative flood damage mitigation and increment (%) after respectively afforesting and sealing the 750 highest ranked pixels in each land use change scenario.**

| Event | Extent | Damages (€) | Damage Mitigation (%) *Afforestation* | Damage Increment (%) *Sealed* |
|---|---|---|---|---|
| *02/2002* | | 566 667 | -41.4 | 1.05 |
| *08/2002* | | 27 515 | -97.3 | 2.80 |
| *2003* | Extent 1 | 49 693 | -99.9 | 0.01 |
| | Extent 2 | 68 405 | -88.0 | 0.12 |
| | Extent 3 | 21 552 | -96.8 | 0.07 |
| | **Total** | **139 650** | **-91.5** | **0.01** |
| *2010* | Extent 1 | 827 122 | -44.3 | 0.77 |
| | Extent 2 | 60 594 | 0.0 | 0.0 |
| | Extent 3 | 366 219 | -16.3 | 0.22 |
| | Extent 4 | 4414 | -67.2 | 0.08 |
| | Extent 5 | 11 382 | -74.1 | 74.8 |
| | Extent 6 | 43 835 | -86.2 | 3.73 |
| | Extent 7 | 100 976 | -22.3 | 0.0 |
| | Extent 8 | 141 813 | 0.0 | 0.0 |
| | **Total** | **1 556 355** | **-39.3** | **1.9** |
| *Maarkebeek* | **Total** | **2 290 187** | **-44.7** | **1.5** |

Figure 10 visualizes, in a spatially explicit manner, where and how much the flood risk was relatively mitigated afforesting 187.5 ha of the most optimal locations for flood volume reduction. The total flood risk mitigation of this afforestation scenario equals a reduction of 57% of the total flood risk (€ 178 252/year), representing an absolute value of € 101 604/year. The highest relative flood risk mitigation was achieved in areas where flood risk was highest, i.e. the built-up, urban areas, by reducing flood depth in these pixels. The relative flood risk increment after implementation of the sealing scenario (Figure 11) equals 0.3%, increasing flood risk with a relatively small increment of € 535/year. Most of this increase was due to the flooding of more pixels, however, similarly to the damages, the flood risk increase is minimal since these pixels are within non-built up area.

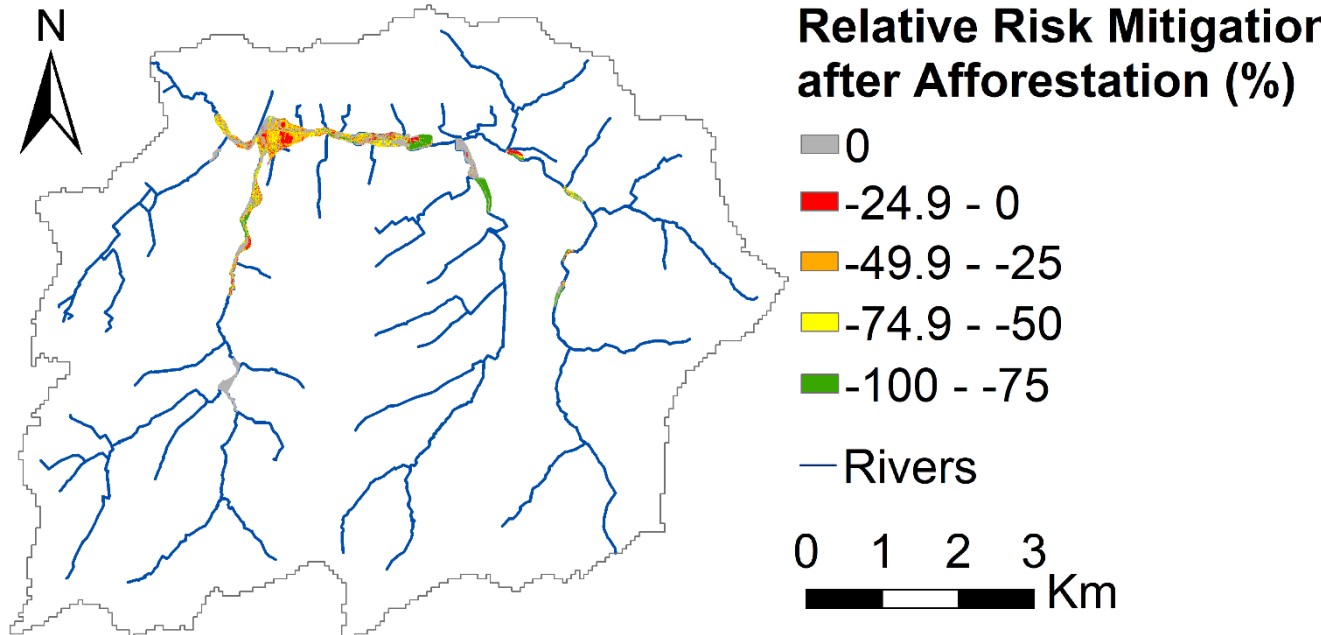

**Figure 10: Relative flood risk mitigation (%) in the Maarkebeek catchment after afforesting the 750 highest ranked pixels in this land use change scenario.**

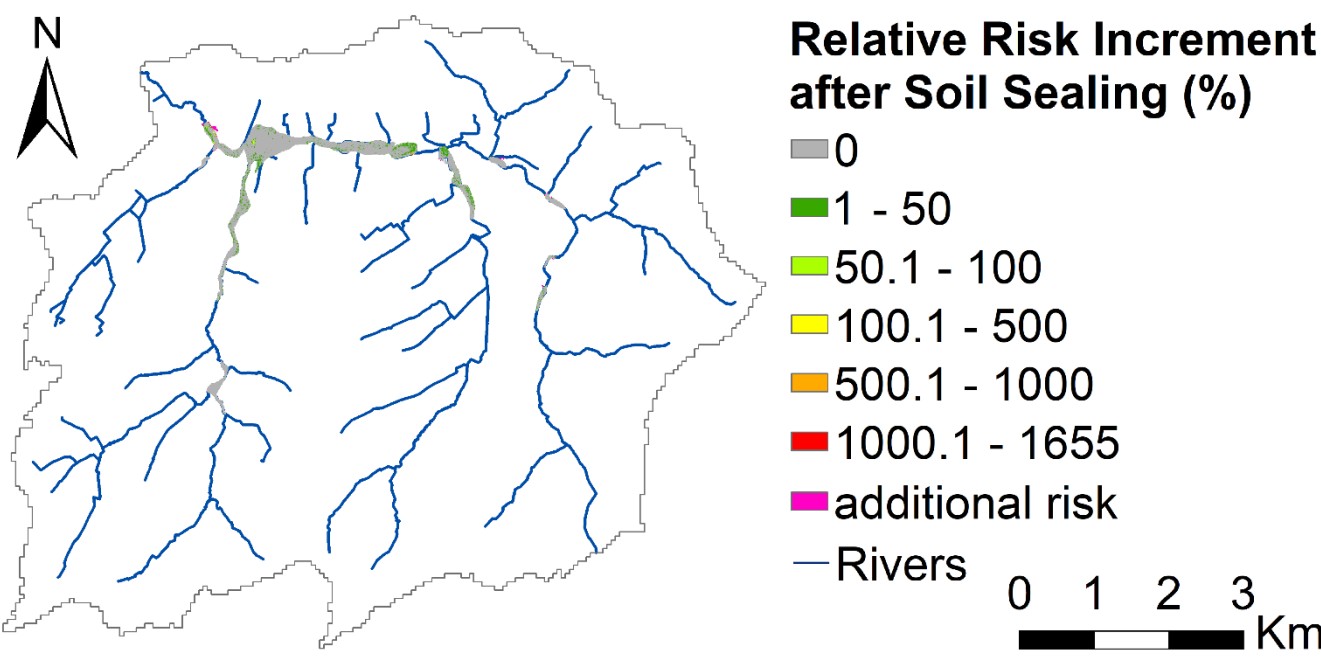

**Figure 11: Relative flood risk increment (%) in the flooded areas in the Maarkebeek catchment after sealing the 750 highest ranked pixels in this land use change scenario. New areas being flooded after soil sealing are depicted as 'additional risk'.**

**4 Discussion**

The results of the comparative flood risk assessment using the proposed framework indicate the potential of identifying optimal locations in catchments for off-site flood damage and risk reduction or minimization of flood risk increment. A limited number of studies have assessed the effect of spatial adaptation measures on flood damages and flood risk. Most notably, Koks et al. (2014) assessed the impact of land-use zoning and compartmentalization on coastal flood risk in Belgium. This study indicated an increase in coastal flood risk without adaptation measures due to socioeconomic developments. Compartmentalization, i.e. upgrading linear elements in the landscape to serve as flood protection, resulted in a higher risk reduction than land-use zoning, i.e. constricting urban development in flood prone areas, which decreased the flood risk by 10 %. The flood risk assessment of soil sealing presented here indicates that constricting soil sealing and urbanization to higher elevations in the catchment results in an overall small relative increment in flood risk of 0.3% or € 535/year, since no additional urban areas are affected by an increase in flood volume. However, this analysis does not take into account urban floods or surcharge of urban drainage systems, which also impact the hydrological response of the catchment leading to an increase in peak discharges (Poelmans et al., 2011).

The relative flood risk reduction resulting from the afforestation scenario is 57% in the Maarkebeek catchment, or € 100 856/year in absolute terms. Figure 11 quantitatively depicts, on a per-pixel basis, where this relative decrease in flood risk is delivered. The absolute flood risk reduction in the Maarkebeek catchment can be compared to the cost associated with the afforestation scenario, estimated based on information provided by E. Van Beek (personal communication, 3/11/2020) and from Van Den Broeck (2019). Saplings costs are approximated at € 1 – 1.5 each, resulting in a cost of € 4000 – 6000/ ha assuming a planting density of 4000 trees/ha. Labour costs are estimated at € 6000, though these costs can be reduced by working with volunteers. The highest cost in afforestation is the acquisition of land, as the price of agricultural land ranges from € 30 000 – 70 000/ha, and averages € 56 595/ha in the province of East Flanders (Federatie van het Notariaat, 2019), wherein the Maarkebeek catchment is situated. Assuming a total afforestation cost of € 67 000/ha in the Maarkebeek, the costs of afforesting 187.5 ha would amount to approximately € 12 500 000. Considering a reduction in flood risk of € 101 604/year, it would therefore take around 125 years for the risk reduction to compensate the costs of afforestation, not taking into account inflation. However, this scenario assumes the acquisition of 187.5 ha of land, constituting 85% of the cost of afforestation. The regional government in Flanders also promotes afforestation among land owners through subsidies, which can total up to € 3250/ha. Under the assumption that a governmental program would sufficient incentives to land owners in the Maarkebeek catchment to afforest 187.5 ha, costing at most € 8750/ha or € 1 640 625 in total, afforestation costs would be compensated by flood risk reduction after approximately 16 years. However, the afforestation scenario used to illustrate the implementation of the comparative risk framework assumes the implementation of a full-grown forest. Consequently, the associated flood risk reduction corresponds to the risk reduction of a full-grown forest, not to a stand of saplings. Hence, afforestation costs would be compensated by the flood risk reduction approximately 16 years after the forest has reached maturity. A more detailed

assessment of the risk reduction pertaining to the different development stages of a forest could be assessed by combining a

forest growth model (e.g. Dalemans et al. (2015)) with a hydrological model (Sutmöller et al., 2011).

The two land use change scenarios on afforestation and soil sealing were used to illustrate the proposed comparative flood damage and risk assessment framework. These land use change scenarios pertain to the optimization procedure, and provide an indication where afforestation and sealing will resp. maximally reduce or minimally increase flood hazard in the flood-

prone areas (Gabriels et al., 2022). These land use changes and their impact on flood hazard are assessed based on a spatial resolution of 50 m by 50 m, however, maximum damage estimates are evaluated based on the 2012 land use dataset with a spatial resolution of 5 by 5 m (AGIV, 2016).

The land use change scenarios were chosen to reflect the spatial planning context in Flanders, a highly urbanized region. Recent spatial planning policy measures aim to reduce flood hazard through the  reduction of soil sealing, the establishment

of green and blue infrastructures and the restoration of natural flood plains (Departement Ruimte Vlaanderen, 2017; VMM, 2019). Especially the reduction of soil sealing is a current point of discussion, centred on the so-called 'building shift'. This policy aims to halt soil sealing in Flanders by removing the building rights on certain plots of land. However, this comes at a high cost, as the loss of these rights by the owners is to be compensated at 100% of the market value by the local authorities (Grommen, 2020). The results of the comparative flood risk framework provide a first indication where to most effectively

allocate the efforts of this building shift by quantifying the flood risk reduction downstream provided by afforestation in the upstream areas. Protecting these locations, and increasing their infiltration capacity through for instance afforestation, will have a higher return on investment in terms of the corresponding reduction in flood risk. The results of the comparative risk assessment thereby point to and quantify the value of green and blue infrastructure and natural flood plains for downstream areas at risk. As such the results of this framework can be integrated in a larger spatial planning assessment of the impact of

land use changes, wherein additional co-benefits of land use changes are also taken into account.

In future applications of the comparative framework, other types of nature-based solutions can be considered, for instance the establishment of natural floodplains. By integrating a more detailed hydrological model, the impact of small-scale landscape elements, such as hedgerows and small ponds, could also be taken into consideration. These small-scale measures, especially when implemented at strategic locations, could reduce the costs associated with large-scale land use changes, which are

relatively high for afforestation, mainly due to the cost of land acquisition. Socio-economic land use dynamics were not taken into account in the implemented land use change scenarios. However, these dynamics will also influence values associated with flood risk mitigation or increment. However, the generic capacity of the presented framework allows the evaluation of any land use change scenario, including more detailed, socio-economic based scenarios.

Validation of flood damage and risk assessments is challenging, as in most cases there is a lack of detailed and consistently updated flood damage databases. Therefore, comparisons between different risk assessments are often used as an alternative validation method (Gerl et al., 2016). Accordingly, the flood risk calculated in this study for the Maarkebeek catchment was

compared to benchmark assessment of economic flood risk performed by the LATIS method, as depicted in Figure 12a. This economic flood risk was determined by combining economic damages of flood events with a return period of 10, 100 and 1000

years. The overall flood risk calculated by LATIS in the Maarkebeek catchment is € 247 255, which is considerably higher than the flood risk of € 178 252/year calculated in this analysis. The difference between the flood risk determined by LATIS and calculated in this analysis is visualized in Figure 12b.  The differences can be explained on the one hand by the larger area at risk of flooding considered in the LATIS too, which is based on modeled flood events with larger return periods. Considering only the pixels at risk of flooding in the presented framework, the LATIS framework estimates flood risk at € 227 139/year.

However, the maximum damage per pixel is higher in the LATIS estimate (€ 9880) than in the presented framework (€ 1265), which is the result of the more extensive economic assessment incorporated in LATIS. The LATIS framework also assesses indirect, internal economic damages, such as clean-up costs, in addition to direct economic damages, which are more comprehensive, including, for instance, damage to vehicles (VMM, 2018). Flood damage assessments typically show a high level of uncertainty in the estimates of maximum damages and in the definition of depth-damage curves (de Moel and Aerts,

2011). Absolute estimates of flood damage therefore have a high level of uncertainty, which is less of an issue when comparing two situations relative to each other, i.e. in the comparison of land us changes, as in the relative risk assessment of the afforestation or soil sealing scenarios (Koks et al., 2014; de Moel and Aerts, 2011).

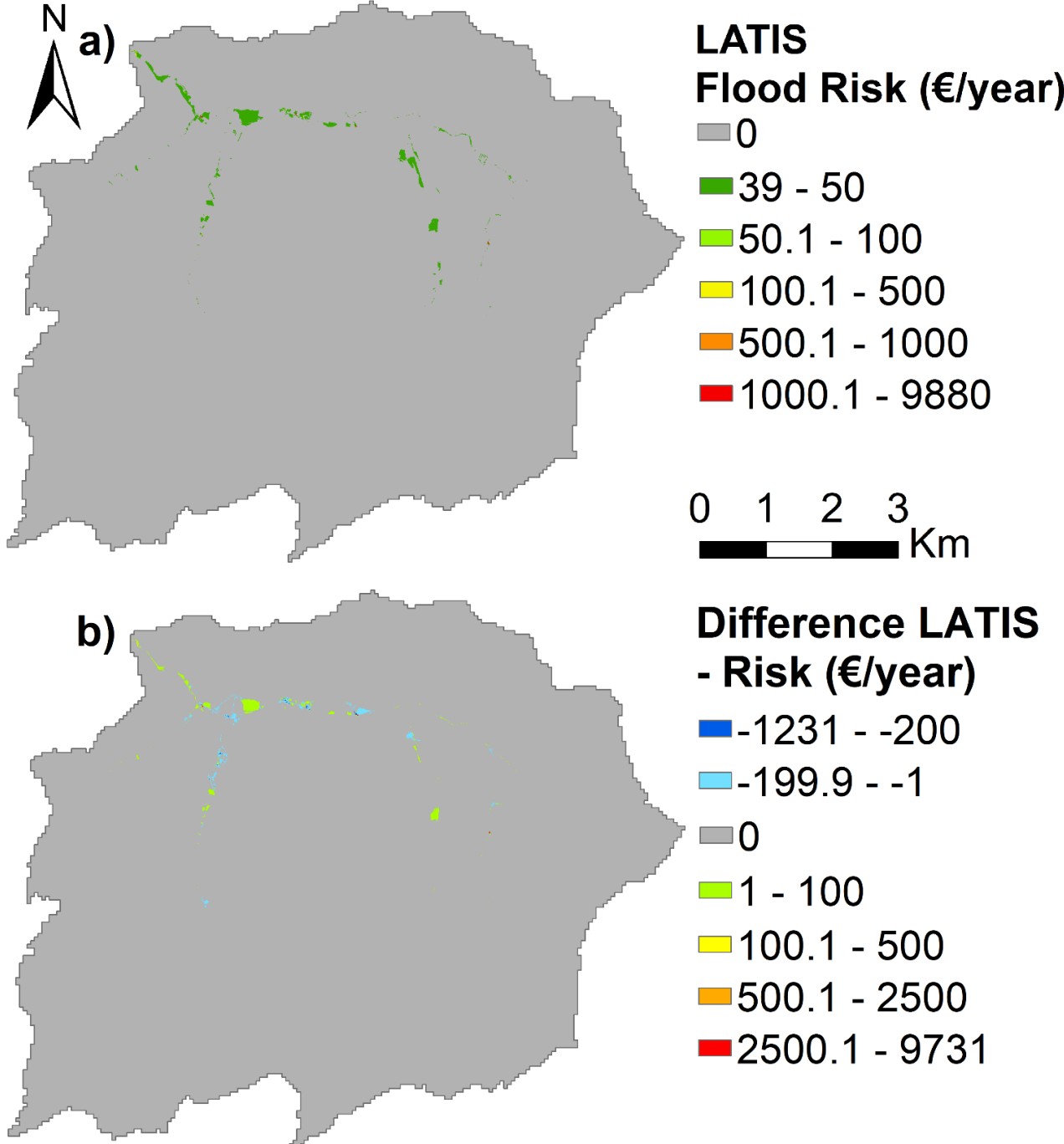

**Figure 12:** (a) Flood risk (€/year per pixel of 25 m²) in the Maarkebeek catchment as calculated by the LATIS tool based on the flood damages determined for flood events with a return period of 10, 100 and 1000 years (adapted from (VMM, 2015)). (source: VMM et al., 2020); (b) Difference in flood risk (€/year per pixel of 25 m²) between the flood risk determined by the LATIS tool and as calculated based on the four observed flood events in the Maarkebeek catchment.


The flood damage assessment only considers direct, tangible flood damage to physical assets, thereby disregarding the impact of flooding on health and the environment. The estimated flood risk reduction or increment associated with land use changes is therefore a reflection of only the direct, tangible economic flood losses. This estimate of direct flood damage does not take into account monetary inflation; the accuracy of this assessment could therefore be increased by adjusting for inflation by using indexed prices to compare housing prices of 2002, 2003 and 2010. Most flood risk assessments are limited to direct, tangible damages (de Moel et al., 2015), as these costs are easy to quantify compared to indirect economic damages (e.g. loss of production of commercial goods for companies situated outside the flooded areas), which would require taking into account complicated supply and delivery chains (Merz et al., 2010). Other risk assessment tools, including LATIS, also provide an indication of social and cultural impacts, together with the loss of life based on the rate of water level rise and flow velocity. In addition, flood duration will also influence total damages, as indicated by flood loss data related to farmland damage (Morris and Brewin, 2014) and residential properties (Mohor et al., 2021). The flood damage assessment also does not take into account adaptive measures, such as Property Level Flood Risk Adaptations (PLFRA), which can be adopted in response to repetitive flooding (Attems et al., 2020; Davids and Thaler, 2021; Joseph et al., 2015). These measures have the capacity to decrease the potential flood damage component of risk, whereas the presented comparative risk framework assesses land use changes as mitigation measures, reducing the hazard component of risk. The flood risk reduction thus reflects the potential risk mitigation value of land use changes in reducing economic flood damage. Depending on the level of implementation of PLFRA, these potential values can overestimate the actual risk mitigation. The implemented flood damage model could be extended and refined to take into account more flood characteristics, allowing a more complete assessment of flood damages. Taking into account adaptive measures, such as PLFRA, would allow the estimated flood risk reduction or increment to reflect more realistic values of risk mitigation or increment.

The presented flood risk assessment assesses flood damage and risk reduction or increment resulting from land use changes based on an event-based rainfall-runoff model, which is straightforward to implement and was also used to derive the two considered land use change scenarios. However, this RR-model is temporally lumped, and as such assesses the hydrological impact of land use changes in terms of runoff volume accumulated during the event (Gabriels et al., 2021). Instead of deriving peak discharge from runoff volume using the rational method (Bingner et al., 2018; Yeo and Guldmann, 2010) and relating the flood peak discharge to flood volume (Mediero et al., 2010), flood volume was directly derived from accumulated runoff through an observed statistical relationship of which the adjusted $R^2$ was 0.76. However, a regional analysis should be performed to assess the applicability of this relationship as in Mediero et al. (2010). A straightforward, conceptual 'bathtub'-model (Teng et al., 2015) was implemented to fill the DEM with the derived flood volumes. This simple method is unable though to accurately simulate inundation in more urbanized settings, where flood risk is highest. Despite the uncertainty related to the RR- and 'bathtub'-model, the uncertainty in flood damage assessments is mostly determined by the implemented estimates of maximum damages and to the depth-damage curves (de Moel and Aerts, 2011). Moreover, this uncertainty is less of an issue in a comparative framework (Koks et al., 2014). Though the implemented methods are thus deemed sufficiently

accurate for a comparative analysis of flood hazard and risk, the genericity of the comparative framework also allows the implementation of more complex hydrological and hydraulic models, able to provide a more detailed modelling of the flood extents and corresponding water depths before and after land use changes.

460

Most flood risk frameworks assess risks based on hypothetical flood events with known return periods, derived from hydrodynamic models encompassing composite hydrographs, which are constructed from extreme value analyses of rainfall-runoff discharge time-series (Kellens et al., 2013; de Moel et al., 2009, 2015; Ward et al., 2011). The impact assessment of land use changes on these hypothetical flood events would therefore require consideration of a long rainfall-runoff time series in order to assess the difference in composite hydrograph and corresponding flood extent. In contrast, the choice was made for the presented framework to implement observed, historical flood events, of which the return periods were estimated based on an analysis of annual maximum discharges. However, the comparison between these observed flood events is restricted, since boundary conditions may have significantly altered between observations (de Moel et al., 2009). Moreover, these historical flood events are characterized by specific meteorological conditions, including rainfall distribution in space and time, which impacts the occurrence of flood extents and thus influence the pixel ranking of the optimization framework. This is reflected in the flood event of August 2002 (Figure 5), where one flood extent was recorded in the east of the Maarkebeek catchment. The historical flood events are thus not as representative to assess flood risk as the hypothetical flood events, and more flood events with a larger range in return periods are required to provide a more comprehensive assessment of flood risk (Ward et al., 2011).

## 5 Conclusion

The presented comparative flood risk assessment framework allows for an estimation of the relative reduction or increase in flood damages and risk due to the implementation of land use changes in the catchment, thereby explicitly taking into account off-site effects of these land use changes in terms of runoff propagation. The comparative flood risk framework was applied in a case study in the Maarkebeek catchment, situated in Flanders, Belgium. Four historical flood events were considered in the risk assessment and their corresponding flood damages and risk were assessed using a flood damage model. Two land use change scenarios of afforesting and sealing 187.5 ha in the catchment were assessed. Comparing flood damages and risk before and after land use change implementation showed a large flood risk mitigation value of afforestation of 57%. This flood risk mitigation value was determined in a spatially explicit manner, depicting which areas benefit the most from afforestation. For the soil sealing scenario, a limited increase of less than 1% in flood risk after soil sealing was modelled. These numeric results are conditioned by the type of scenario implemented in the framework, in this case including the 750 pixels which after afforestation contribute maximally to reducing the flood volume resp. the 750 pixels which after sealing contribute minimally to increasing the flood volume.

Apart from its obvious strengths for assessing the flood risk impact of land use changes, this framework also has limitations,

some inherent to flood damage estimation, such as the uncertainty in maximum damage estimates and depth-damage curves, and some specific to this assessment, as it is based on observed flood events rather than hypothetical flood events with known return periods. Moreover, it derives flood volumes from runoff volume accumulation based on an empirical relationship, which should be further established using regional analyses. Despite these limitations, the framework provides the possibility for computationally efficient and spatially explicit assessments of the flood mitigation value or relative risk increment associated

with potential land use changes. The generic framework can be applied for studying the effectiveness and efficiency of a variety of nature-based solutions, whereby other, more detailed hydrological and hydraulic models can be integrated to further refine the estimated values of flood risk reduction or increment. As such, this framework can be used as an explorative tool in spatial planning processes related to flood risk management.

**Author contribution**

All three authors conceptualized the methodology presented in this paper. Karen Gabriels developed the model code and performed the simulations under supervision from Patrick Willems and Jos Van Orshoven. The manuscript was prepared by Karen Gabriels with significant contributions from both co-authors, Jos Van Orshoven and Patrick Willems.

**Competing interests**

The authors declare that they have no conflict of interest.

**Acknowledgements**

This research was made possible by the research foundation FWO (Fonds WetenschappelijkOnderzoek) through a PhD scholarship (Grant SB/1S01217N) to Karen Gabriels.

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
