# Peer review of "A comparative flood damage and risk impact assessment of land use changes"

_Natural Hazards and Earth System Sciences, 2021_

## Referee Comment (RC1)

Review:

Title: A comparative flood damage and risk impact assessment of land use changes
Author(s): Karen Gabriels et al.
MS No.: nhess-2021-51
MS type: Research article

| location | Text | Issue to correct / refine |
|---|---|---|
| Abstract | These measures include the establishment of land use types with a high (e.g. forest patches) or low (e.g. sealed surfaces) water retention and infiltration capacity at strategic locations in the catchment. | Use of sealed surfaces is a bit unusual and might imply the 'opposite of a nature-based solution'? It needs more explanation of how this has influenced the hydrology. Are you modelling urbanisation? |
| Abstract | Rainfall runoff model | Can you name the model here please and describe it in more detail |
|  | Sealing scenario | Needs more explanation – are you suggesting artificial surfaces? |
| Line 34 | *Finally, the flood risk is determined by combining the flood damages caused by flood events with different return periods in a weighted summation.* | I think this definition should be more scientific, e.g.: Risk can be quantified in terms of an average annual damage, by weighting the computed impacts for design events with their respective annual exceedance probabilities. |
| Para lines 40-50 |  | Reads well |
| Line 51 | …LATIS…. | Would it be better to introduce LATIS alongside your model and state the differences from the outset – especially the differences in assumptions on e.g. indirect damages. LATIS gets used and results presented but we don't really understand the background to it here. |
| Line 82 | *Using a flood damage model, flood damages were assessed from four flood events occurring in the Maarkebeek basin between 2000 and 2016* | This flood damage model needs to be defined and more details provided. What were the antecedent conditions for each event? Soil moisture will strongly influence the assumptions on runoff generation |
| Line 83 | *The overall flood risk was determined by combining the flood damages of the four events with their respective probability of occurrence.* | Please specify the assumption made at low return periods for the onset of flooding – do you assume onset at the median flood or somewhere between that and the minimum flood hazard used? What probability events were used? Please specify here. |
| Line 86 | *spatially explicit rainfall-runoff (RR) model, calculating the runoff volume accumulated in each pixel after a rainfall event* | Please specify the model used!!! |
| Line 95 | Return periods | Define in relation to annual exceedance probability and also specify which ones! |
| Line 97 | *Consequently, an empirical relationship between observed flood volumes and modeled runoff volume accumulation is established to* | I think I understand this, but perhaps some curves of volume versus runoff should be shown to illustrate? Did you build a look-up table to relate the |

| | | |
|---|---|---|
| | *determine the flood volumes after land use changes.* | volume on the floodplain to runoff (volume? Peak flow?) for the available flood hazards? You could refer to figure 5 later. |
| Line 99 | *Based on these modeled flood volumes, a DEM is progressively filled and corresponding water depths are thus determined.* | One issue here is what happens if water in reality not going to reach a certain pixel until depths overcome an embankment? Do all depressions fill up simultaneously? I think this is very approximate |
| Line 110 | Deriving depths and volumes | This technique seems too approximate – some measure of uncertainty in the level might be useful – or a sensitivity analysis |
| Figure 1 | diagram | I think the diagram could be explained better. I guess the delta is the estimated change in impacts. Perhaps an example of the statistical relationship would help such as a curve of depth versus runoff accumulation? |
| 117 | CN-based | CN? Needs more explanation please |
| 118 | *This CN-based RR-model propagates the runoff through the watershed, thereby continuously assessing downstream re-infiltration using the Manning's equation.* | The Manning's equation is for open-channel flow – please explain tis better – how does it help assess the re-infiltration are you talking about a difference? |
| 126 | equation | Ok I have also used a similar relationship so good to see this here. |
| 140 | Figure 2 | Interesting that the damage factor is so high for shallow flooding of roads – why is this? Is it relating to disruption losses? |
| 149 | Household damages | What was the average max damage per unit area used for residential? Please provide as you provide this for other receptors and later in the paper |
| 164 | Weighted summation | Not sure about the 'double counting' here – you are weighting the damages with the return period – or annual exceedance probability – it is not removing double counting? |
| 170 | Equation 4 | The important factor here is your lowest return period modelled – as it sets the limit of what we know about the onset of flooding. What is your smallest RP modelled hazard? Was it 10 years? |
| 193 | Interpolation | This is a better explanation /summary than earlier |
| 199 | Residential damages | Worth using earlier to give a feel for the range |
| 219 | This procedure ranks pixels based on (i) where in the upstream area of the flooded zones afforestation maximally reduces the runoff accumulation in these zones, and | I do not understand how you have moderated the upstream accumulations in the modelled land use change – how is this represented in the model. This is really important to the credibility. Are you assuming the top 750 pixels don't contribute anymore or is there |

| | | |
|---|---|---|
| | | some sort of fractional reduction? On what evidence is it based? The modelled change leads to a BIG reduction in damages, so needs fully explaining.

 Also the afforestation must surely have a different impact on runoff depending on the soil moisture – which depends on the antecedent rainfall . I think you need to state your assumptions or model more conditions, and also which hydrological processes you are representing. |
| 220 | Prioritising pixels | I think the uncertainties will be high in this approach – It would be good to understand the sensitivity of the outcomes: for example - the errors in damages that could be incurred due to +/- 0.1m error in the water surface level |
| 236 | determine | Replace use of this word with estimate |
| 243 | error | There are 2 formatting errors |
| 255 | inflicted | Replace with 'incurred' |
| Figure 5 | discussion | Discuss the impact of the lower return period damages mainly being underestimated by the regression model compared to the data. These will all be weighted more strongly. Perhaps a two stage relationship is needed? Again, sensitivity to this would help understand the decisions that could be made. |
| Figure 6 | 5m*5m impact cells | If you are using 5m then may be a flow accumulation grid using 5m DTM would have been more appropriate / compatible (instead of 50m)? |
| Line 277 | Comparative damages | These reductions in damages are very large and of concern – would woodland really have such a big impact?

 – what antecedent conditions do you assume in the 'model' for different storms? |
| 285 | Sealing scenario | The change is not nearly so great |
| 353 | comparison | It is good to compare with other estimates |
| 357/8 | Comparison with LATIS | if indirect damages are being assessed in one model but not the other then it might be possible to just use a factor to correct and allow a better comparison. |
| | | |
| 360 | uncertainties | …but you could help define the uncertainties better with more sensitivity analysis here |
| Figure 11 | LATIS reporting | I'm not sure why the LATIS reporting and outputs are shown here as it's not |

| | | been used in the main study? Is this just for comparison – in which case a side-by-side plot might be more useful with your method or an overlay. |
|---|---|---|
| Line 373 + 382 | Use of regression | See comments about two stage or exponential regression, plus this section should include information summarising antecedent soil moisture for the calibration events – are they different? How do they vary seasonally? You finally mention boundary conditions in the last section – I think this is very important and may account for some of the scatter etc. |
| 385/conclusions | First sentence | The framework does allow for this comparison but I think the uncertainties must be very high, and the predicted reductions in flood risk seem very high. |
| 394 | 57% reduction | I haven't seen values this high – you need explain what mechanisms in hydrology can help with this – are you including:
• Increased infiltration
• Increased soil storage
• Increased transmissivity in soil profile
• Increased wet-canopy evaporation
• Increased friction
I think you need to explore how the changes you've imposed in the model are justified in relation to hydrological processes |
| | | |
| | | |

---

## Author Response (AR1)

Dear editor,

We thank you for the opportunity to revise and improve our manuscript according to the comments made by the reviewers. Below we have detailed our point-by-point response to the comments with references to the line number where adjustments have been made in the manuscript.

Thank you for your time and consideration in reviewing the revisions we have made.

Kind regards,

Karen Gabriels, Patrick Willems and Jos Van Orshoven

**COMMENT REVIEWER 1**

*I think this is an interesting paper with a good data-based approach, but it needs more work on explanations, understanding of hydrological processes being modelled, the impact of antecedent soil moisture and some analysis of sensitivity to modelling assumptions such as interpolations of water surface on the outcomes. The reduction in impacts for afforestation is very large.*

Thank you for your time and consideration in reviewing our manuscript. We agree that the discussion of our manuscript can further be extended. We further elaborate on how we propose to adjust our manuscript according to your remarks.

The derivation, calibration and validation of hydrological rainfall-runoff model is further detailed in a recently published paper:

Gabriels, K., Willems, P., & Van Orshoven, J. (2021). Performance evaluation of spatially distributed, CN-based rainfall-runoff model configurations for implementation in spatial land use optimization analyses. *Journal of Hydrology*, *602*, 126872.
https://doi.org/https://doi.org/10.1016/j.jhydrol.2021.126872

This paper also includes an extensive discussion of the limitations of this model. We propose to provide a reference to this paper in our manuscript and to strengthen the discussion of our manuscript to include the implications of the most relevant modelling processes and assumptions, including the adjustments of the model to antecedent soil moisture conditions. The interpolations of water surfaces will impact the estimate of water depth and therefore also the flood risk assessments. However, this uncertainty relates to both the reference situation, before land use change, and after land use changes have been implemented. The uncertainty related to the water depth does therefore pose less of an issue in the relative assessment of both situations, as proposed in this framework. We will clarify and extend on this in the discussion of our manuscript (L434–441 in revised manuscript).

The reduction in impacts for the land use change scenarios are difficult to validate, as validation data are lacking. To provide some indication on the accuracy of the flood risk assessment, we provided a comparison with the flood risk model LATIS (Beullens et al., 2017):

Beullens, J., Broidioi, S., De Sutter, R., De Maeyer, P., Verwaest, T., & Mostaert, F. (2017). *Ontwikkeling LATIS 4 Deelrapport bis: Actualisatie basiskaarten en schadewaarden. Versie 3.0. WL Rapporten, 13_159_7.* Universiteit Gent, Antea Group, Waterbouwkundig Laboratorium: Antwerpen.

This model has been used in Flanders to provide a benchmark economic flood risk assessment. The discussion on the uncertainty related to the comparative risk assessment will be strengthened by adding additional information regarding the assumptions made in the land use change scenarios (L229–234 in revised manuscript). For instance, the assumption is made that a full-grown forest is implemented, which will lead to the overestimation of the impact of afforestation (L354–359 in revised manuscript). In addition, the combination of a limited number of flood events with a limited number of return periods does also influence the impact assessment of the land use changes, with a larger number of events and return periods leading to a more accurate assessment (L449–454 in revised manuscript).

**COMMENT REVIEWER 2**

*I have three main queries under flood damage estimates: residential damage is likely overestimated, the role of property-level flood risk adaptations (PLFRA), and the critical role of duration.*

Thank you for your time and consideration in reviewing our manuscript. Your comments definitely provide an interesting perspective on our manuscript. Below we provide a response to your remarks and indicate how we will adjust and improve our manuscript accordingly.

*For the estimates of flood damage to residential buildings – the value of a home includes the land, the services to the land, e.g. sewer, water, electricity, and the property itself. Therefore, the cost of refits/rebuilding after a flood is some fraction of the value of the property. Estimates could be based on insurance pay-outs or other data.*

*For the analysis there are repeat floods in the same areas, yet the estimates of damage use the same formula – however, we might expect households, farmers, etc to implement PLFRA. Indeed, insurers may require such PLFRA. Some articles on PLFRA are:*
*https://wires.onlinelibrary.wiley.com/doi/full/10.1002/wat2.1404 and*
*https://www.witpress.com/elibrary/sse-volumes/5/3/995*

*Duration is discussed earlier in the paper, but Figure 3 is just extent and Figure 6 is depth and damage estimates. Duration is critical in terms of damage costs to farmland and to residential properties – including to intangible costs. See here for estimates of the role of duration on farmland damage,*
*https://onlinelibrary.wiley.com/doi/epdf/10.1111/jfr3.12041*

The maximum damage values in our manuscript provide an estimation of the values of the properties exposed to the flood. These estimates are based on the average housing prices in the different municipalities. We hereby follow the approach as implemented in the LATIS model, the benchmark model in Flanders for flood risk assessments (Beullens et al., 2017):

Beullens, J., Broidioi, S., De Sutter, R., De Maeyer, P., Verwaest, T., & Mostaert, F. (2017). *Ontwikkeling LATIS 4 Deelrapport bis: Actualisatie basiskaarten en schadewaarden. Versie 3.0. WL Rapporten, 13_159_7.* Universiteit Gent, Antea Group, Waterbouwkundig Laboratorium: Antwerpen.

The damage after flooding is then derived from the expert-based water depth-damage curves, which are also applied in LATIS. These curves detail the fraction of the value of the property representing the flood damage for a given water depth. We implemented this approach, as there is a lack of consistent, complete and spatially distributed data on insured flood damages in Flanders. Indeed, this economic flood risk assessment does not consider the resilience to flood damage, which can be increased through

the implementation of property-level flood risk adaptations. It also does not take into account the duration of flooding, though we acknowledge that this will also influence the total damage resulting from the flood event. The economic assessment in our manuscript is limited to the direct, tangible flood damage. Therefore, the ecosystem insurance value estimates only relate to these direct, tangible damages.

We will make the assumptions related to the damage estimates, and the underlying reasons for these assumptions, more explicit in the introduction of our manuscript (L54–55 and L82–85 in revised manuscript). We will add to the discussion the limitations of this direct, tangible economic flood risk assessment and discuss the implementation of PLFRA and the impact of flood duration with the inclusion of the interesting references you have provided and other relevant publications (e.g. https://www.cogitatiopress.com/urbanplanning/article/view/4246) (L404–424 in revised manuscript).

*I understand how you estimated the afforestation and sealing scenarios, but I wondered if there is a threshold in either scenario? There must be threshold effects with increased sealing of the uplands and with undermining the benefits delivered by natural upstream areas, i.e. once they are opened for development the natural area will be under greater pressure for development. Surely these areas contribute a lot to the provision of flood regulation ecosystem services?*

Our land use scenarios do not take into account threshold effects. The land use scenarios pertain to the optimization exercise, and provide an indication where afforestation and sealing will resp. maximally reduce or minimally increase flood hazard in the flood-prone areas. Socio-economic land use dynamics are thus not considered in the implemented afforestation and sealing scenarios. However, these dynamics will surely play a role and influence the ecosystem insurance value. Our generic comparative flood risk assessment framework allows any land use change scenario to be evaluated, also more detailed, socio-economic based scenarios. As such, the generic capacity of the framework provides perspectives for research on questions of the type the reviewer mentions.

*The takeaway that it is OK to build in the uplands near forest patches and afforestation is primarily around rivers seems somewhat counterintuitive and these unexpected results are not explicitly discussed and need to be. Are there other more realistic scenarios that could be generated, i.e. a scenario that pays attention to development and conservation planning in this catchment? This might include afforestation that also occurs in the upland and development that occurs in already residential areas or near these areas. Yet another scenario could try to estimate the mitigation fraction provided by upland conserved/forested areas by modelling the removal and/or partial removal of these areas.*

The message of our manuscript is not that it is OK to build in the uplands and afforestation should only occur in the lowlands, but rather that building in the uplands leads to a lesser increase of peak discharge and flood volumes than building in the lowlands, and that afforestation of the riparian zones leads to a larger reduction of peak discharge and flood volumes than afforestation of the uplands. We can add further context to the results of the optimization analysis, thereby also highlighting the limitations in allocating sealing and afforestation with the single objective to reduce flood hazard. We will further nuance the findings of the optimization exercise and add references to other research reflecting these findings (e.g. https://hess.copernicus.org/articles/14/325/2010/) (L236–243 in revised manuscript).

As the land use change scenarios were selected to reflect the spatial planning context in Flanders, a highly urbanized region, we can also refer to recent policy measures implemented in Flanders, protecting and

restoring wetlands in river valleys with a view to reduce flood hazard. As mentioned, the proposed framework allows ecosystem insurance value to be derived for any land use change scenario. As the focus of this manuscript is more on the proposed risk assessment framework rather than on the land use change scenarios, analysis of additional land use change scenarios, although perfectly possible, would be beyond the scope of this manuscript. The land use change scenarios we addressed are related to Gabriels et al. (2022):

Gabriels, K., Willems, P., & Van Orshoven, J. (2022). An iterative runoff propagation approach to identify priority locations for land cover change minimizing downstream river flood hazard. *Landscape and Urban Planning*, *218, 104262. https://doi.org/10.1016/j.landurbplan.2021.104262*

In future research the implementation of other types of solutions can be considered, for instance the implementation of flood control reservoirs or the establishment of floodplains. The latter would require a more detailed hydrological model, to enable the consideration of small-scale landscape elements. These measures could indeed reduce associated costs. We propose to add a section to the discussion related to the perspectives on future research offered by our generic flood risk assessment framework (L361–382 in revised manuscript).

*As you note the costs of afforestation are high, another scenario could assess other NBS, i.e. the best places for floodplain floodwater storage. An advantage of such a scenario is that the storage of floodwaters would be temporary which might reduce associated costs.*

*The afforestation mitigation outcomes seem very high – are these % reductions similar to those found in other research?*

The validation of land use change scenarios is challenging, and little comparable research is available assessing the impact of land use change scenarios on flood risk. A comparison is made with the findings of Koks et al. (2014) (https://link.springer.com/article/10.1007/s10113-013-0514-7). We will also include a reference to our recently published paper, outlining the rainfall-runoff model implemented in the flood risk assessment framework:

Gabriels, K., Willems, P., & Van Orshoven, J. (2021). Performance evaluation of spatially distributed, CN-based rainfall-runoff model configurations for implementation in spatial land use optimization analyses. *Journal of Hydrology*, *602*, 126872. https://doi.org/https://doi.org/10.1016/j.jhydrol.2021.126872

This paper provides additional insights into the uncertainty related to the hydrological modelling. In order to provide some validation for the flood risk assessment itself, a comparison was made between the estimated flood risk with the original land use and the bench-line economic flood risk estimates provided by the model LATIS (Beullens et al., 2017).

*It would be useful for Readers who do not know the area to have the main towns located on the figures and to have an overall view of the area, i.e. an indication of where the rural areas, farmland, residential areas are.*

We agree and will add a land use map of the study area to our manuscript (p. 8 in revised manuscript).

*Many of the references seem older or perhaps it is the absence of more recent references that is an issue.*

We have tried to include the most relevant references to our research. We could search for more recent references; however, limited research has been done to assess the impact of land use/ecosystem changes on flood risk. Our manuscript presents a novel perspective on the ecosystem insurance value, which could provide an interesting incentive for future research in this important field of study.

*Word choice:*

- *Around line 33 – elements – perhaps 'assets'. This word is repeated several times.*
- *Around line 124 – analogue – perhaps 'an analogue to…' or 'analogously' or 'similar to' – again this word is repeated several times.*

We agree and will adjust our manuscript accordingly.

---

## Author Response (AR2)

Dear editor and reviewers,

We thank you for your time and consideration in reviewing our revised manuscript. We have taken the minor revisions suggested by reviewer 2 into full consideration and integrated them into our manuscript. Below we have detailed our point-by-point response to the minor revisions with references to the line number where adjustments have been made in the manuscript.

Thank you for your continued interest and effort in reviewing our manuscript.

Kind regards and best wishes for the new year,

Karen Gabriels, Patrick Willems and Jos Van Orshoven

**COMMENT REVIEWER 2**

Thank you for your time and consideration in reviewing our revised manuscript. Below we have outlined point-by-point the adjustments we have made according to your suggestions in order to further improve the quality of our manuscript.

*As correctly stated somewhere toward the end of the discussion: the numbers are surrounded by large uncertainties. Nevertheless, I find from the abstract until the conclusion EUR values up to the single unit. I suggest replacing them by kEUR to focus more on the order of magnitude of the results instead of the exact values.*

The numbers mentioned in the manuscript abstract have been adjusted to represent the values in kEUR (L. 15–16 in the revised manuscript).

*Abbreviate corporate author names when used in references. For some of them it is done correctly (e.g. EEA in line 20), other like VMM are sometimes used, sometimes not and for example Agentschap Informatie Vlaanderen, Nationaal Geografisch Instituut, Agentschap Innoveren en Ondernemen are not. The full names, especially when in combinations break the reading flow and therefore the good understanding of these sentences.*

Following your suggestion, we abbreviated all corporate author names in our manuscript and in the references.

*Cartography can be improved for almost all figures. Select a line to delineate the catchment and then have a white background as a starting point. In addition, select different colours and legend classes to improve the readability and relevance of the maps included (More detailed comments below for individual maps).*

The depiction of the figures is based on the catchment delineation of the Maarkebeek. We did not alter the scale of this depiction to a smaller part of the catchment, since the flood events are distributed over the larger part of the Maarkebeek catchment. We have attempted to improve the readability of the figures in our manuscript according to the detailed suggestions provided for each individual map, providing a white background and adjusted scales. These adjustments are described for each figure in the response on the detailed comments.

*Important information is the level of detail of the land use maps: these have pixels of 50*50m² resulting in 100 pixels of the DEM having the same land use class. That's an additional source of uncertainty in the results not mentioned.*

The general land use dataset depicted in Figure 3 has a resolution of 50 m (L. 183–184). This general land use dataset is implemented in the RR-model and thus used in the optimization framework to

assess land use changes. This information is added in L. 186. It is, however, important to note that the maximum damage estimates are based on the land use dataset of 2012 with 5 m resolution (L. 216–217). As such, the DEM and the land use dataset used to derive maximum damage values do have the same resolution of 5 m. This is now clarified in our manuscript in L. 374–376.

*The paper referred to (Gabriels et al. 2022) is not yet available to understand some work. However, after enlarging the maps on screen, it looks as the selection of the 750 pixels is based on the RR-model CN parameters only and does not take into account aspects like connectivity to already existing forest plots or to each other. Same for the sealed areas that can occur in the middle of agricultural land and are not necessarily connected to existing urban areas. Is this assumption correct? Although it will probably be expressed in the other paper, some additional details would be welcome to understand what is done here.*

The paper of Gabriels et al. (2022) is now available online, the DOI is provided in the references and it can be found on https://www.sciencedirect.com/science/article/abs/pii/S0169204621002255. We agree that some context is lacking to interpret the resulting, selected pixels for afforestation and sealing. The RR-model is implemented in the optimization framework, thus the spatial connectivity between pixels is explicitly taken into account. This is added in L. 125 with regards to the RR-model and repeated in L. 236–237 in the context of the optimization procedure. Further details are also provided in L. 238–245 to provide further context to the spatial distribution of the selected pixels: 'In each of the two flood events in 2002, only one flood extent was observed; the most downstream pixel in this extent, i.e. the outlet, was consequently used as point of interest (POI) in the optimization and pixels were ranked based on the change in runoff volume accumulation at this POI. In the flood events in 2003 and 2010, respectively three and eight flood extents were observed. These extents' outlets were considered the POIs in the optimization and the pixels were ranked based on the combined changes in runoff accumulation at these pixels, weighted according to the observed flood damages in each flood extent. The four optimization results, one for each of the flood events, were summed to obtain one ranking for each land use change, thereby weighting the standardized pixel ranks according to the flood hazard, i.e. as the corresponding flood damages are weighted in Eq. (5).'

*Detailed remarks:*

- *lines 19-20: There's an indicator for 1980-2019 as well, the EEA reference you give is over a decade old (https://www.eea.europa.eu/ims/economic-losses-from-climate-related). Europe is defined here as EEA member countries or EU-27 (eventually plus UK).*

We opted originally for the older reference, since this relates detailed figures concerning floods, i.e. hydrological events. The more recent indicator relates information regarding all extreme weather events, which also includes economic damages from storms (meteorological events) and droughts (climatological events). The data source of the indicator is not available for further data analysis, however, from the data table we extracted and summed the damages related to hydrological events, resulting in a total of 147 billion EUR. We updated the reference in our manuscript and added the total sum in L. 19-20.

- *lines 55-57: in particular for agricultural damages, the historic damages were compared with LATIS results. A limited comparison with damage records in the (at the time federal) Rampenfonds were done as well in addition to using NL and UK enquiries.*

This information has been added to our manuscript in L. 56–57.

- *lines 83-84: in a newly added sentence you describe the work as similar to LATIS, while in the discussion a lot of differences are mentioned. I suggest to remove the newly added sentence.*

Following your suggestion, we have removed this sentence from our manuscript.

- *line 87: the four events are nowhere defined above and only become clear when looking at Figure 4 several pages later.*

We provided this information in the introduction to describe the general outline of the manuscript. We acknowledge that this information is provided too soon in the manuscript, thus we removed the reference to four specific flood events from the introduction and describe the methodology in a more general way (L. 83–85).

- *lines 156-157: while not applied in the study for VMM in 2006, LATIS can also distinguish maximum damage estimates for Flemish 'landbouwstreken'. Especially in areas with a lot of vegetables, maximum prices differ significantly from maize or wheat.*

We added this information to our manuscript in L. 162–164.

- *Figure 3: somewhere in the text above Figure 3, describe the pixel size of the land use map (see general remark).*

The pixel size of Figure 3 is detailed in L. 183-184.

- *Figure 4b: Select different colours for the flooded areas or consider reversing the colour scale for the DEM. For the 2002 maps: what about overlapping areas flooded in both events? Which of the 2 events is mapped on top of the other? Why is this not a fourth individual map? What do the results show in the next maps: the sum of both, the average, one overlaying the other event?*

A different colour was selected to depict the flooded areas in order to provide a larger contrast with the underlying DEM. The flood events in February and August 2002 each consist of one recorded flood extent. These extents do not overlap, thus, these extents are depicted in the same figure, with different colours to distinguish the extent belonging resp. to the event in February and August. This has been clarified in the text (L. 198–200) and in the caption of Figure 4.

- *Figure 5: delineate the area with a line and leave the background (not selected) white to increase readability. Sealing can get a different colour, for example red to increase readability.*

The background of Figure 5 is changed to white and sealing is depicted in a red colour. Though no areas were delineated with line features, we feel these adjustments do increase the readability of the figure.

- *Figure 7a: I only see green. As pixels with high(er) water depth are probably located close to the rivers, the information is invisible. A continuous linear scale is maybe not the best choice.*

The pixels with high water depths are indeed localized close to the rivers. A linear scale indeed does not well display the distribution of water depths. Water depths are now classified in four relevant classes, leading to more colour variation in the figures.

- *Figure 7b: there's a yellow legend class and a yellow background. Background can be removed. The legend classes result in only green colours being visible. Consider a different scale and add the maximum value to the highest legend class to indicate the range instead.*

The background of Figure 7b is changed to white. The class boundaries have been changed and more classes are added to allow for a better differentiation in the figures.

- *Figure 8: 0 is not bigger than 1265 and should be placed at the other end of the legend (or removed). I see some dashes of red and yellow, but a different legend choice (class boundaries, eventually adding 1 class) would indicate much better areas of attention (and in later figures where the biggest benefits are).*

The legend has been ordered in ascending order and class boundaries have been adjusted, adding an additional class, in order to allow more differentiation between areas with different risk values.

- *Figure 9 a and b: a yellow background with a yellow legend class do not match. Background can be white. In 9b, 0 should be placed at the other end of the legend.*

The background has been changed to white and the legend has been reordered in ascending order.

- *Figure 11: 0 at the wrong end of the legend.*

The background has been changed to white and the legend has been reordered in ascending order.

- *Lines 371-375: it is important to mention the price of land, as this is often forgotten in nature-based solutions. The same for labour forces. Both of them can be partially mitigated (volunteers, volunteering schemes). For the subsidies: these have a cost as well, so they don't replace the cost of land acquisition. And in addition, how likely is it that the 750 pixels selected will be the ones where the afforestation measures are applied? Planting trees is a good idea, but maybe flood reduction is not the primary benefit. I miss topics like ancillary effects (co-benefits) in a discussion that is lengthy and to the point.*

The high costs associated with afforestation are related in L. 353-358, mentioning the labour costs as well as the cost of the land acquisition. In L. 390–392 the impact of small-scale landscape elements is mentioned, which are associated with lower implementation costs.
The presented comparative flood risk assessment framework was illustrated using the land use change scenario aimed at reducing flood risk hazard in the catchment. However, it can be applied on any land use change scenario (L. 396–397). The framework is aimed at quantifying the flood risk mitigation or increment associated with land use changes. Other co-benefits associated with these land use changes are not taken into consideration. However, the results of this framework can be part of a larger assessment of the impact of land use changes in spatial planning. This has been added to the discussion in L 388–389.

- *Figure 12: 0 at wrong end of the legend. Can there in addition to map 12 (which will become 12a) a map 12b presenting the differences in between the 2 results visually?*

The legend classes have been reordered in ascending order and additional classes were defined. A difference map between LATIS and the risk estimated by the flood risk assessment framework was also added. This figure is also described in L399–401 and the caption of the figure was adjusted.

- *Lines 442 and following: Suddenly, a new element is introduced being the flood insurance value. In my opinion it does not contribute much to the overall paper as it neglects many aspects on how insurance mechanisms work, insurance premiums, compensation values. The discussion has enough substance without this aspect.*

Indeed, this new element would require more elaboration on the mechanism of insurances. Following your suggestion, the references to the flood insurance value have been removed from our manuscript.